# Oblivious Unlearning by Learning: Machine Unlearning Without Exposing Erased Data

## Abstract

Machine unlearning enables users to remove the influence of their data from trained models, thus protecting their privacy. However, it is paradoxical that most unlearning methods require users first to upload their to-be-removed data to machine learning servers and notify the servers of their unlearning intentions to prepare appropriate unlearning methods. Both unlearned data and unlearning intentions are sensitive user information. Exposing this information to the server for unlearning operations conflicts with the privacy protection goal. In this paper, we investigate the challenge of implementing unlearning without exposing erased data and unlearning intentions to the server. We propose an Oblivious Unlearning by Learning (OUbL) approach to address this privacy-preserving machine unlearning problem. In OUbL, the users construct a new dataset with synthesized unlearning noise, ensuring that once the server incrementally updates the model using the original learning algorithm based on this dataset, it can implement unlearning. The server does not need to perform any tailored unlearning operation and remains unaware that the constructed samples are for unlearning. As a result, the process is oblivious to the server regarding unlearning intentions. Additionally, by transforming the original erased data into unlearning noise and distributing this noise across numerous auxiliary samples, our approach protects the privacy of the unlearned data while effectively implementing unlearning. The effectiveness of the proposed OUbL method is evaluated through extensive experiments on three representative datasets across various model architectures and four mainstream unlearning benchmarks. The results demonstrate the significant superiority of OUbL over the state-of-the-art privacy-preserving unlearning benchmarks in terms of both privacy protection and unlearning effectiveness.

## 1 Introduction

Machine unlearning enables users to exercise the right to remove the influence of specific data samples from trained machine learning (ML) models, thereby protecting user privacy. Paradoxically, while the goal of machine unlearning is to protect user privacy, most unlearning methods necessitate that users upload their specified data to the ML server as a prerequisite for executing the unlearning process (Bourtoule et al., 2021; Warnecke et al., 2023). Additionally, users must inform the server that the uploaded data are intended for unlearning purposes, enabling the server to prepare and execute the corresponding unlearning methods and operations (Thudi et al., 2022; Hu et al., 2024b).

However, these requirements expose the privacy of users' unlearning data and intentions, rendering existing unlearning methods impractical in privacy-sensitive scenarios. In privacy-preserving ML contexts, such as those described in (Cao et al., 2021; Bonawitz et al., 2017; Sun et al., 2022; Naseri et al., 2022), the server is restricted from accessing individuals' data due to privacy concerns. Moreover, modifying the original learning algorithms is challenging in these contexts, as most ML models are trained using secure mechanisms like federated learning (FL) (Cao et al., 2021; Naseri et al., 2022) or secure multi-party computation (MPC) (Mohassel & Zhang, 2017; Knott et al., 2021). Additionally, even if the erased samples are protected, exposing unlearning intentions can allow adversaries to conduct inversion attacks targeted at unlearning updates (Hu et al., 2024a; Chen et al., 2021; Zhang et al., 2023). Exposing erased data and unlearning intentions to ML servers for unlearning contradicts the privacy-preserving frameworks' requirements (Naseri et al., 2022; Bonawitz et al., 2017; Cao et al., 2021) and undermines the fundamental purpose of the right to

be forgotten (Liu et al., 2022b; Dang, 2021). Therefore, a privacy-preserving machine unlearning service is crucial and necessary, and we explore the following question: "*Is it possible to achieve machine unlearning without revealing users' erased data and unlearning intentions to the server?*"

**Motivation.** Machine learning models can be regarded as mappings of the training data (Goodfellow et al., 2016; Shalev-Shwartz & Ben-David, 2014). Consequently, altering the training data can change the model's performance, such as poisoning and backdooring methods (Lin et al., 2020; Zeng et al., 2023; Tramèr et al., 2022; Liu et al., 2022a), data "unlearnable" methods (Fu et al., 2022; Sandoval-Segura et al., 2022), and adversarial attacks (Kang et al., 2024; Madry et al., 2018; Zhou et al., 2023). Inspired by these works, we investigate the construction of a synthetic dataset with unlearning noise that ensures the unlearning effect when the server updates the model based on the constructed dataset. Since the model update uses only the original learning algorithms, users do not need to inform the server to prepare specific unlearning operations, thereby protecting the privacy of unlearning intentions.

In this paper, we begin by reformulating the privacy-protection unlearning problem as an *oblivious unlearning by learning problem* to investigate the research question. We propose an Oblivious Unlearning by Learning (OUbL) strategy to address this problem. The goal is to synthesize a new dataset that includes both clean samples and unlearning noise-injected auxiliary samples. The unlearning effect is achieved when the server updates the model using the original learning algorithm on the synthesized dataset. The implementation of OUbL hinges on two key aspects: (1) precisely estimating the unlearning model update as the target for unlearning noise generation, using only the information of the unlearning user, and (2) designing an efficient method to generate the unlearning noise for the auxiliary dataset to achieve the desired unlearning effect. Specifically, we first propose an efficient unlearning update estimation method based on Hessian-vector products, which samples only the data of the unlearning user, ensuring that users can calculate it efficiently by themselves. Second, we generate the unlearning noise through gradient matching, i.e., finding the noise-injected data with gradients update similar to the estimated unlearning update. We propose an unlearning noise descent method to efficiently synthesize the noise, treating the noise matrix as an input layer and fixing the entire model, thereby only calculating the gradient for the noise layer for the update.

We conducted extensive experiments on three representative datasets and compared our method with four mainstream unlearning benchmarks to evaluate both privacy protection and unlearning effectiveness. To assess privacy protection, we performed unlearning inversion attacks (Hu et al., 2024a; Zhang et al., 2023) to reconstruct the erased samples across different unlearning methods, comparing their reconstruction similarity to demonstrate the privacy protection effect. A lower reconstruction similarity indicates better privacy protection. For evaluating unlearning effectiveness, we employed a prevalent data removal verification method, MIB (Hu et al., 2022). The experimental results demonstrate that OUbL offers superior privacy protection and unlearning effectiveness compared to existing privacy-preserving unlearning methods (Wang et al., 2023; Liu et al., 2022b). In comparison with state-of-the-art unlearning methods without privacy protection (Bourtoule et al., 2021; Nguyen et al., 2020), OUbL incurs only a slight trade-off in model utility.

Our contributions are summarized as follows:

- To the best of our knowledge, this paper is the *first* to identify the privacy threats posed by the exposure of both unlearning intentions and unlearned data during machine unlearning processes. It highlights the paradox of the existing unlearning methods that require unlearning users to upload raw data and inform the server to prepare customized unlearning algorithms, which conflicts with the original privacy-protection goal of the right to be forgotten.
- We propose an OUbL approach to protect unlearned data and unlearning intentions during machine unlearning processes. OUbL contributes a precise unlearning updates estimation method and an efficient unlearning noise generation method to ensure the unlearning effect when the server updates the model using the original learning algorithm based on the constructed dataset.
- We conducted extensive experiments to compare OUbL with exact and approximate unlearning methods, with and without privacy protection. The results validate OUbL's superiority in terms of privacy protection and unlearning effectiveness over existing privacy-preserving methods, with only a slight trade-off in model utility compared to unlearning methods without privacy protection.
- The source code and the artifact of the OUbL is released at `https://anonymous.4open.science/r/OUL-55F6`, which creates a new tool for protecting the privacy of erased data and unlearning intentions during machine unlearning processes.

## 2 PRELIMINARY AND PROBLEM STATEMENT

To facilitate the understanding of the privacy-preserving machine unlearning problem, we first introduce the mainstream process of unlearning. A detailed discussion about the "Related Work" of machine unlearning is presented in Appendix A.

**Machine Unlearning.** The unlearning process typically includes the following phases: (1) The server trained a model with parameters $\theta_o$ derived from dataset $D$. (2) The user uploads the dataset $D_u$ for which they request unlearning to the server, indicating the data to be erased from the model. (3) Upon receiving the unlearning request, the server executes an unlearning algorithm $\mathcal{U}$ to remove the contributions of $D_u$ from $\theta_o$, resulting in an unlearned model $\theta_{D \setminus D_u}$, also denoted as $\theta_u$.

Note that this is a standard machine unlearning process without privacy protection, which exposes both the unlearning intentions and the erased data to the server in phases (2) and (3). To protect the privacy of erased samples and unlearning intentions, it is necessary to modify phases (2) and (3). These modifications should ensure that unlearning can be implemented without exposing $D_u$ to the server and without informing the server of the unlearning intention, needing to eliminate the dependence on specified unlearning algorithms.

One primary challenge is the need to eliminate reliance on tailored unlearning methods from the server, thereby avoiding the exposure of unlearning intentions. Given that the server is aware of the original learning algorithm $\mathcal{A}$ and that model updates are a reasonable requirement in real-world scenarios (Kirkpatrick et al., 2017; Wu et al., 2019; Wang et al., 2022), we pose the question: *Can we achieve unlearning through incremental learning to prevent the server from detecting unlearning intentions?* In addition to protecting unlearning intentions by solely executing incremental learning, a privacy-preserving mechanism $\mathcal{C}$ is necessary to safeguard the erased data. Furthermore, the scheme should not come at the cost of significant model utility degradation. Therefore, we formulate the privacy-preserving unlearning problem into an *oblivious unlearning by learning problem* as follows.

**Problem Statement** (Oblivious unlearning by learning). *Suppose the ML server has an original trained model $\theta_o$, trained using algorithm $\mathcal{A}$ on dataset $D$. Let the unlearning user possess the unlearned dataset $D_u = (X_u, Y_u)$, where $D_u \subset D$. Oblivious unlearning by learning aims to **(1)** protect the privacy of the unlearned data $D_u$ by designing a mechanism $\mathcal{C}(D_a, D_u) \to D_a^p$ that conceals the erased data as unlearning noise on users' new updating auxiliary dataset $D_a$, and **(2)** protect the privacy of unlearning intentions by achieving the unlearning effect through incrementally updating the model $\theta_o$ using the original learning algorithm $\mathcal{A}$ on $D_a^p$. To preserve model functionality, the incrementally updated model should attain a similar model utility as traditional unlearning algorithms $\mathcal{U}$, i.e.,*

$$\mathcal{U}(\theta_o, D_u, D) \approx \mathcal{A}(\theta_o, \mathcal{C}(D_a, D_u)). \tag{1}$$

To solve the Eq. (1), since the learning algorithm $\mathcal{A}$ and the unlearned data are fixed, our focus shifts to designing the dataset construction mechanism $\mathcal{C}$. This mechanism must effectively protect the privacy of $D_u$ and ensure the desired unlearning effect using the constructed dataset $\mathcal{C}(D_a, D_u)$ during model updating.

## 3 OBLIVIOUS UNLEARNING BY LEARNING (OUbL)

### 3.1 BASIC IDEA AND OVERVIEW OF OUbL

Assume we have an unlearned model $\theta_u$ and the original trained model $\theta_o$. The unlearning update is given by $\Delta\theta_{D_u} = \theta_u - \theta_o$. When we update a model based on a new dataset, such as $D_a^p = \mathcal{C}(D_a, D_u)$, we have $\theta \leftarrow \theta_o - \nabla\ell(D_a^p; \theta_o)$. To achieve unlearning based on the incremental learning update, we must ensure that:

$$\theta_u = \theta_o - \frac{1}{P} \sum_{(x,y) \in D_a^p} \nabla\ell((x,y); \theta_o), \quad \text{where} \quad \frac{1}{P} \sum_{(x,y) \in D_a^p} \nabla\ell((x,y); \theta_o) = -\Delta\theta_{D_u}, \tag{2}$$

and $P$ is the size of $D_a^p$. If we can construct a dataset $D_a^p$ meets the requirement of Eq. (2), we can achieve the oblivious machine unlearning by learning, guaranteeing (a) the erased data is hidden from the server as it has not been used in the update, and (b) the unlearning intention is hidden to the server as there is no unlearning request, just normal model update. However, to achieve the oblivious

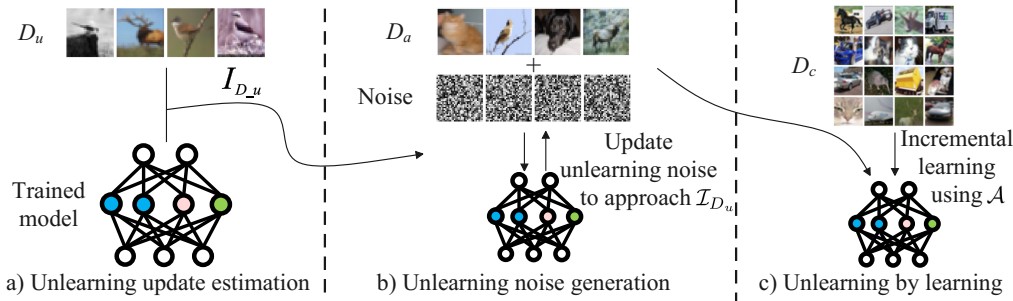

a) Unlearning update estimation     b) Unlearning noise generation     c) Unlearning by learning

Figure 1: The main process of OUbL includes three components. First, the unlearning user estimates the influence $\mathcal{I}_{D_u}$ of unlearning the erased data from the trained model. Second, the user customizes the noise for an auxiliary dataset such that incrementally training the model based on the synthesized auxiliary dataset $D_a^p$ can effectively unlearn the erased data. Third, the unlearning user prepares a clean dataset $D_c$ to preserve the model utility and uploads both the clean and synthesized auxiliary datasets to the server for incremental learning, thereby achieving the unlearning effect.

unlearning effect, there are two main challenges: (1) it is hard to achieve a precise unlearning model update before we execute the unlearning procedure with the above restriction; (2) even if we achieve the unlearning update, it is challenging to construct the dataset so that the server can incrementally update based on the constructed dataset to achieve the unlearning effect.

**Overview of OUbL.** We propose an approach called OUbL to overcome the aforementioned challenges. The main process of OUbL, illustrated in Figure 1, comprises three key components: unlearning update estimation, generating unlearning noise to synthesize auxiliary data, and unlearning by learning training with utility compensation.

## 3.2 UNLEARNING UPDATE ESTIMATION

Many existing approximate unlearning methods first estimate the unlearning influence and then reduce the estimated unlearning influence from the trained model for unlearning (Guo et al., 2020; Sekhari et al., 2021; Liu et al., 2022b). A representative unlearning influence estimation method involves the use of a Hessian matrix-based approximation, which can be described as follows:

$$\Delta\theta_{D_u} = \theta_{D\backslash D_u} - \theta_o \quad \simeq \mathcal{I}^{(1)}(D_u) = \frac{1}{n-m} H_{\theta_o}^{-1} \sum_{(x_u, y_u) \in D_u} \nabla\ell((x_u, y_u); \theta_o), \tag{3}$$

where $H_{\theta_o}^{-1}$ denotes the inverse of the Hessian matrix evaluated at $\theta_o$ on the dataset $D\backslash D_u$, $n$ is the size of $D$, and $m$ is the size of $D_u$. $\nabla\ell$ denotes the gradients of the learning algorithm with loss function $\ell$, and $\mathcal{I}^{(1)}(D_u)$ denotes the estimated first-order influence of the unlearning data $D_u$. Eq. (3) is commonly used in (Guo et al., 2020; Sekhari et al., 2021; Liu et al., 2022b).

However, studies (Guo et al., 2020; Sekhari et al., 2021; Liu et al., 2022b) calculate Eq. (3) with the assistance of the entire remaining dataset, which is prohibited in privacy-concerning scenarios. In our setting, the unlearning user cannot access the entire training dataset $D$; they can only access their own dataset. Moreover, forming and inverting the Hessian of the empirical risk $H_{\theta_o}^{-1}$ requires $O(np^2 + p^3)$ operations, as the original dataset contains $n$ samples and $\theta_o \in \mathbb{R}^p$. This computation is too expensive for large neural networks.

We propose an efficient unlearning update estimation (EUUE) method to overcome the aforementioned challenges by using Hessian-vector products (HVPs) to directly approximate the unlearning update $H_{\theta_o}^{-1} \sum_{(x_u, y_u) \in D_u} \nabla\ell((x_u, y_u); \theta_o)$. This idea is inspired by (Agarwal et al., 2016; Koh & Liang, 2017). It requires only the assistance of the erased samples and a few clean training samples of the unlearning user, ensuring that the user can calculate it. For clarity, we drop the $\theta_o$ subscript. We employ an estimator that samples a single point per iteration for a fast Hessian matrix calculation. Let $H_j^{-1} \stackrel{\text{def}}{=} \sum_{i=0}^{j} (I - H)^i$ be the first $j$ terms in the Taylor expansion of $H^{-1}$, where $I$ is the identity matrix. We can rewrite this recursively as $H_j^{-1} = I + (I - H)H_{j-1}^{-1}$. From the validity of the Taylor expansion, $H_j^{-1} \to H^{-1}$ as $j \to \infty$. We can substitute the full $H$ with a

draw from any unbiased estimator of $H$ to form $\tilde{H}_j^{-1}$ at each iteration. We have $\mathbb{E}[\tilde{H}_j^{-1}] \rightarrow H^{-1}$ as $\mathbb{E}[\tilde{H}_j^{-1}] = H_j^{-1}$. The detailed process of EUUE is presented in Algorithm 2 in Appendix B.

## 3.3 Generating Unlearning Noise for Auxiliary Dataset

Having achieved the estimated unlearning update $\Delta\theta_{D_u}$ from the above process, we can reformulate the condition of Eq. (2) for noise synthesis. Instead of directly synthesizing the dataset, we generate noise for a normal update dataset. This approach avoids generating samples that significantly differ from the original data, which could harm the model's utility. Assume we now have an auxiliary dataset $D_a : (X_a, Y_a)$. We add the noise $\Delta^p$ to $D_a$, resulting in $D_a^p = (X_a + \Delta^p, Y_a)$, ensuring that the model update is similar to the update for unlearning. Specifically, we need to satisfy $\frac{1}{P}\sum_{(x,y) \in D_a^p} \nabla\ell((x,y); \theta_o) + \Delta\theta_{D_u} = 0$ according to Eq. (2), which can be reformulated as follows for finding suitable noise.

$$\min_{\Delta^p} \|(\frac{1}{P}\sum_{(x_i, y_i) \in D_a} \nabla\ell((x_i + \Delta_i^p, y_i); \theta_o) + \Delta\theta_{D_u}\|, \quad \text{s.t. } \Delta^p \in \arg\min_{\theta} \frac{1}{N}\sum_{i \leq N} \ell((x_i + \Delta_i^p, y_i); \theta),$$

(4)

where $N$ is the size of $D \setminus D_u \cup D_a$, and the value of $\Delta\theta_{D_u}$ is estimated according to Eq. (3).

**Noise Synthesis by Gradient Matching.** Our objective is to find noise $\Delta^p$ such that, when the model is trained on the noise-synthesized auxiliary samples $D_a^p$, it minimizes the two losses in Eq. (4), thus making the model unlearn the erased samples. However, directly solving Eq. (4) is computationally intractable due to the bilevel nature of the optimization objective. Instead, one may implicitly minimize the unlearning update by finding suitable $\Delta^p$ such that for any model parameter $\theta$, the following condition is satisfied:

$$\Delta\theta_{D_u} \approx -\frac{1}{P}\sum_{(x_i, y_i) \in D_a} \nabla_\theta\ell((x_i + \Delta_i^p, y_i); \theta).$$

(5)

If we can enforce Eq. (5) to hold for any $\theta$ during training, then the gradient steps that minimize the training loss on the synthesized auxiliary data will also minimize the unlearning target, satisfying $\frac{1}{P}\sum_{(x,y) \in D_a^p} \nabla\ell((x,y); \theta_o) + \Delta\theta_{D_u} = 0$. Unfortunately, calculating $\Delta^p$ that satisfies Eq. (5) is also intractable as it is required to hold for all values of $\theta$. In our setting, the unlearning user cannot access $\theta$ for samples in the remaining dataset $D \setminus D_u$. One possible solution proposed in (Geiping et al., 2021; Di et al., 2022) is to relax Eq. (5) to be satisfied for a fixed model — the model obtained by training on the original dataset. We assume a well-trained model $\theta_o$ before unlearning and fix it during unlearning noise generation. Then, we can minimize the loss based on the cosine similarity between the two gradients as:

$$\phi(\Delta^p, \theta_o) = 1 - \frac{\langle \Delta\theta_{D_u}, -\frac{1}{P}\sum_{i=1}^{P} \nabla_\theta\ell((x_i + \Delta_i^p, y_i); \theta_o)\rangle}{\|\Delta\theta_{D_u}\| \cdot \| -\frac{1}{P}\sum_{i=1}^{P} \nabla_\theta\ell((x_i + \Delta_i^p, y_i); \theta_o)\|}.$$

(6)

Geiping et al. (Geiping et al., 2021) use $R$ restarts, usually $R \leq 10$, to increase the robustness of noise synthesis. Using this scheme, we can also find suitable noise for unlearning. However, it is not always effective because we cannot always achieve satisfactory random noise within 10 restarts. To address this issue, we propose an unlearning noise descent strategy.

---

**Algorithm 1:** Unlearning Noise Descent (UND)

---

**Input:** Trained model $\theta_o$, unlearning update estimate $\Delta\theta_{D_u}$, auxiliary dataset $D_a$
**Output:** The synthesized data with the unlearning noise, $D_a^p = (X_a + \Delta^p, Y_a)$
1  **procedure UND** $(\theta_o, \Delta\theta_{D_u}, D_a)$**:**
2     $\Delta_1^p \leftarrow \mathcal{N}(0,1)$    ▷ Initialize unlearning noise.
3     **for** $i \leftarrow 1$ **to** $n$ **do**
4         $X_{a,i}^p \leftarrow X_a + \Delta_i^p$   ▷ Add the noise to data.
5         $\nabla\theta_{o,i} \leftarrow \nabla_\theta\ell((X_{a,i}^p, Y_a); \theta_{o,i})$   ▷ Compute gradients.
6         $\phi_i \leftarrow \text{Sim}(\Delta\theta_{D_u}, \nabla\theta_{o,i})$   ▷ Compute similarity using Eq. (6).
7         $\Delta_{i+1}^p \leftarrow \Delta_i^p - \eta\nabla_{\Delta^p}(\phi_i)$   ▷ Update noise to match gradients.
8     **return** $D_{a,p} = (X_a + \Delta_{n+1}^p, Y_a)$

---

**Unlearning Noise Descent.** Algorithm 1 synthesizes unlearning noise to create a perturbed dataset $D_{a,p} = (X_a + \Delta^p, Y_a)$. Firstly, we generate a noise matrix $\Delta^p$ as shown in line 1 of Algorithm 1 and treat it as parameters that could be updated during optimization. Then, during optimization, we fix the trained model parameters $\theta_o$ and add the noise to the auxiliary data $D_a$ as the input to the model (line 4). We calculate the gradients of the noise-synthesized data based on the current model point but do not update the model (line 5). Moreover, we calculate the minimization loss according to Eq. (6) (line 6). With this loss, we can use the gradient descent method for both the model and the unlearning noise matrix, but we only update the noise matrix $\Delta^p$ while keeping the model $\theta_o$ fixed (line 7). After a few rounds of iteration, we can synthesize sufficient unlearning noise to the auxiliary data to achieve the unlearning effect.

## 3.4 Oblivious Unlearning Guarantee and Utility Preservation

**Oblivious Unlearning Guarantee by Incremental Learning.** Can gradient alignment cause model to converge to a model with low unlearning update approaching loss? To simplify presentation, we denote the unlearning update approaching loss $\mathcal{L}_{unl}$ and incremental loss $\mathcal{L}_{inc}$ of Eq. (4) as

$$\mathcal{L}_{unl}(\theta_o) =: \|(\frac{1}{P} \sum_{(x_i,y_i) \in D_a} \nabla \ell((x_i + \Delta_i^p, y_i); \theta_o) + \Delta\theta_{D_u}\|, \tag{7}$$

$$\mathcal{L}_{inc}(\theta_o) =: \frac{1}{P} \sum_{i \le P} \ell((x_i + \Delta_i^p, y_i); \theta_o). \tag{8}$$

Additionally, recall that $1 - \phi(\Delta^p, \theta_o)$ measures the cosine similarity between the unlearning update and the incremental training update in Eq. (6). By adapting a classical result of Zoutendijk, the Theorem 3.2 in (Nocedal & Wright, 2006), we can elucidate why the unlearning effect can be accomplished by merely performing standard incremental training on the synthesized dataset.

**Proposition 1** (Unlearning Descent by Learning). *Let $\mathcal{L}_{unl}(\theta_o)$ be bounded below and have a Lipschitz continuous gradient with constant $L > 0$ and assume that the ML server incrementally trains the model by gradient descent with step sizes $\alpha_k$, i.e. $\theta_o^{k+1} = \theta_o^k - \alpha_k \nabla \mathcal{L}_{inc}(\theta_o^k)$. If the gradient descent steps $\alpha_k > 0$ satisfy*

$$\alpha_k L < \beta(1 - \phi(\Delta^p, \theta_o^k)) \frac{\|\nabla\mathcal{L}_{inc}(\theta_o^k)\|}{\|\nabla\mathcal{L}_{unl}(\theta_o^k)\|}, \tag{9}$$

*for some fixed $0 < \beta < 1$, then $\mathcal{L}_{unl}(\theta_o^{k+1}) < \mathcal{L}_{unl}(\theta_o^k)$. If in addition $\exists \epsilon > 0$, $k_0$ so that $\forall k \ge k_0$, $\phi(\Delta^p, \theta_o^k) < 1 - \epsilon$, then*

$$\lim_{k \to \infty} \|\nabla\mathcal{L}_{unl}(\theta_o^k)\| \to 0. \tag{10}$$

See supplementary material in Appendix D

**Model Utility Preservation.** Based on the above processes, the unlearning effect can be achieved by simply having the server update the model $\theta$ on the synthesized $D_a^p$ using the learning algorithm $\mathcal{A}$. However, updating only based on the unlearning noise-injected data will compromise the model utility to some extent. To compensate for the model utility degradation, the unlearning user can mix some clean samples $D_c$ to $D_a^p$ as the final uploading dataset $D_{up} = D_a^p \cup D_c$. Following the empirical risk minimization (ERM) loss function, we can rewrite the incremental learning loss as:

$$\mathcal{L}_{D_a^p \cup D_c}(\theta) =: \frac{1}{P + C} \sum_{(x,y) \in D_a^p \cup D_c} \ell((x, y); \theta), \tag{11}$$

where $P$ is the size of $D_a^p$ and $C$ is the size of $D_c$. Since $D_c$ are clean samples, training on $D_a^p \cup D_c$ will mitigate the accuracy degradation, meanwhile with more clean samples in $D_c$ to cover the unlearning noise-synthesized $D_a^p$ will also enhance the privacy protection for erased data.

## 4 Experiments

### 4.1 Settings

**Datasets and Models.** We have conducted experiments on three widely adopted public datasets: MNIST (Deng, 2012), CIFAR10 (Krizhevsky et al., 2009), and CelebA (Liu et al., 2018), offering a

range of objective categories with varying levels of learning complexity. The statistics of all datasets are listed and introduced in Appendix E.1. We select three model architectures of different sizes in our experiments: a 5-layer multi-layer perceptron (MLP) connected by ReLU on MNIST, a ResNet-18 on CIFAR10 and a 7-layer convolutional neural network (CNN) on CelebA.

**Metric.** To effectively evaluate privacy protection, we propose a **reconstruction similarity** metric, which calculates the cosine similarity between the erased samples and the reconstructed samples by the recovering attacking based on the unlearning update (Hu et al., 2024a; Salem et al., 2020; Zhang et al., 2023). These attacks are the most serious type of attack that aims to recover the erased information, and higher reconstruction similarity means more information is leaked. We also conduct reconstruction attacks on scenarios with and without unlearning intentions to clearly illustrate the importance of unlearning intentions in Section 4.4.

To effectively evaluate the unlearning effectiveness, we refer to prevalent data removal verification methods (Hu et al., 2022; Guo et al., 2023), adding the backdoor patches to the unlearning samples $D_{u,b} \leftarrow (X_u + patches, Y_u + target)$ for the original model training. Then, we execute the unlearning methods to unlearn the backdoor-marked samples $D_{u,b}$. After unlearning, we test the **backdoor accuracy** on the marked-unlearning samples $D_{u,b}$ to evaluate the unlearning effect. Moreover, since OUbL implements unlearning by training the model to approach the estimated unlearning update, we propose an **unlearning update similarity** metric to evaluate the unlearning effect achieved by OUbL, which is calculated using the cosine similarity between the estimated unlearning update and the truly unlearning update after conducting OUbL on the trained model.

Lastly, we use **model accuracy** to evaluate the model utility and functionality preservation, and we evaluate the efficiency of unlearning methods based on their **running time**. All the metrics are summarized in Appendix E.2.

**Compared Machine Unlearning Benchmarks.** To comprehensively compare OUbL with existing machine unlearning methods, we consider the comparison with both no-privacy protection unlearning methods and privacy protection methods. Specifically, we choose one exact unlearning method, SISA (Bourtoule et al., 2021), and one approximate unlearning method, VBU (Nguyen et al., 2020), as centralized no-privacy protection methods. We choose two state-of-the-art federated unlearning methods, BFU (Wang et al., 2023) and Hessian matrix-based federated unlearning (HBFU) (Liu et al., 2022b), as erased data privacy protection methods. We choose BFU and HBFU rather than other federated unlearning methods such as (Su & Li, 2023; Lin et al., 2024) because BFU and HBFU achieve the best unlearning effect. In contrast, (Su & Li, 2023; Lin et al., 2024) focus more on improving unlearning efficiency, which compromises unlearning effectiveness to some extent. We briefly summarize these methods in Appendix E.3.

## 4.2 PRIVACY PROTECTION EVALUATIONS

In this section, we evaluate the privacy protection of OUbL and the compared benchmarks. To clearly demonstrate the privacy protection for the erased samples, we not only show the evaluation of the federated unlearning method, we also design a baseline by directly adding local differential privacy (LDP) noise to the erased samples, $D_{u,\epsilon} = (X_u + \texttt{LDP}(\epsilon), Y_u)$, and implementing unlearning using VBU, which is called VBU-LDP. We choose VBU to implement the baseline privacy-protection unlearning with LDP noise because VBU can implement unlearning only based on the erased samples. Other unlearning methods require the assistance of the remaining dataset besides the erased samples; combining LDP with these methods will inject too much noise and significantly compromise the model's utility. The privacy protection evaluation results on three datasets are presented in Figure 2.

**Setup.** The variable of this experiment is $\epsilon$, which controls the injected LDP noise. By comparing the performance of OUbL with the VBU-LDP with different $\epsilon$, we can directly observe which privacy protection level the OUbL achieved. In this experiment, we set the unlearning samples rate (USR), $USR = 1\%$ for all unlearning methods, and constructed samples rate $CSR = 1\%$ and auxiliary samples rate $ASR = 1\%$ for OUbL.

**Privacy Protection Evaluated by Reconstruction Attacks.** The first column in Figure 2 shows the reconstruction similarity, attacking by (Hu et al., 2024a; Zhang et al., 2023) to recover the erased samples of different unlearning methods. Note that the privacy protection of OUbL when informing

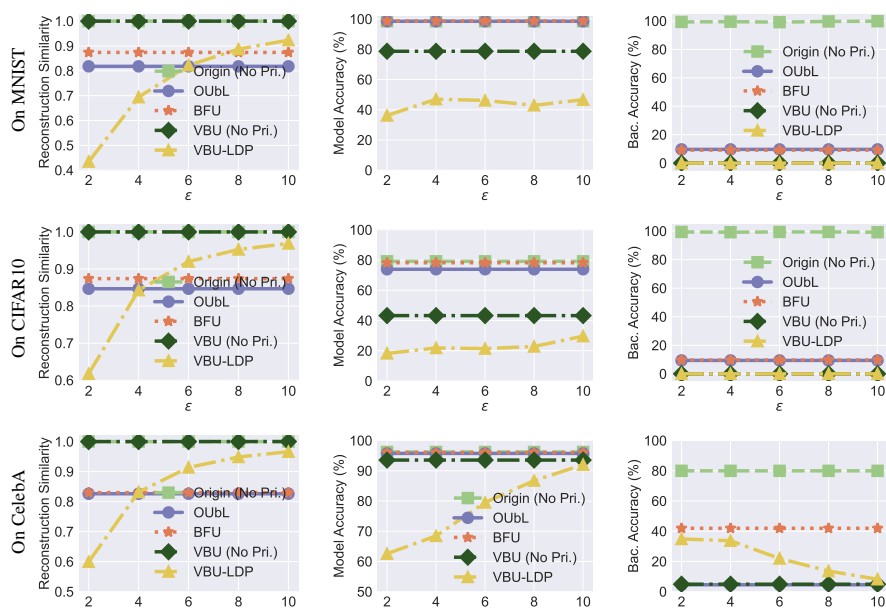

Figure 2: Comparisons between OUbL and LDP-based Methods

the server of unlearning intentions would be similar to the privacy protection of FL-based methods. Since the reconstruction similarity of OUbL with unlearning intentions information overlaps with BFU, we omit showing the performance of OUbL with unlearning intentions. Instead, we compare OUbL with and without knowing unlearning intentions in Figure 5 in Section 4.4.

By hiding the unlearning intentions, OUbL can always achieve better privacy protection than BFU. The attack is conducted based on the model updates on the server side, where the server of BFU knows the unlearning intentions as it needs to inform other users to retrain, and the server of OUbL does not know the unlearning intentions as it treats the OUbL process as a usual updating. OUbL has a huge reconstruction similarity decrease than BFU on MNIST and CIFAR10 and a slight decrease than BFU on CelebA.

Moreover, we compare OUbL with VBU-LDP to illustrate the privacy effect. OUbL achieves privacy protection like $\epsilon = 6$ LDP privacy protection of VBU-LDP on MNIST, similar to $\epsilon = 4$ LDP privacy protection of VBU-LDP on CIFAR10 and $\epsilon = 4$ LDP privacy protection on CelebA.

**Trade-off between Privacy Protection and Unlearning Efficacy.** With good privacy protection for the erased samples, OUbL simultaneously achieves much better model utility preservation than VBU-LDP and a much better data removal effect than BFU. For VBU-LDP, it is obvious that noise injection significantly decreases the model accuracy. There is a huge gap between VBU and VBU-LDP on the second column in Figure 2, and smaller $\epsilon$ decreases the model utility worse. Although BFU has not compromised the model accuracy with the assistance of normal federated users, it sometimes fails to unlearn the samples, such as the last figure on CelebA in Figure 2.

### 4.3 UNLEARNING EFFICACY AND EFFICIENCY EVALUATIONS

**Setup.** We first illustrate the incremental training details of OUbL on MNIST, CIFAR10 , and CelebA to demonstrate unlearning effectiveness as shown in Figure 3. We also compare OUbL with two centralized unlearning methods, SISA and VBU, and with two federated unlearning methods, BFU and HBFU, in Figure 4. In this experiment, we conduct the evaluation for multi-sample unlearning, where $USR = 1\%$. We set the $CSR = 1\%$ and $ASR = 1\%$.

**Results of Unlearning Efficacy and Efficiency.** From the unlearning efficacy perspective, Figure 3 definitely shows the unlearning effectiveness of incremental training on synthesized data (OUbL) compared with clean data. The backdoor accuracy on the unlearned dataset of OUbL drops obviously as the incremental training on synthesized data continues, which is confirmed in all three datasets. In Figure 4, OUbL achieves similar model accuracy and backdoor accuracy as SISA, a representative exact unlearning method without erased data privacy preservation. Although VBU

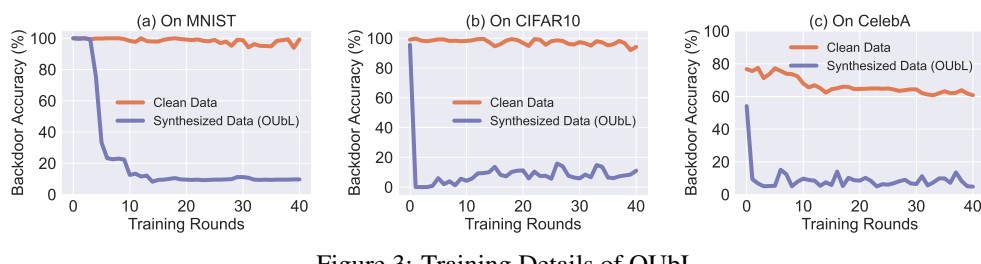

Figure 3: Training Details of OUbL

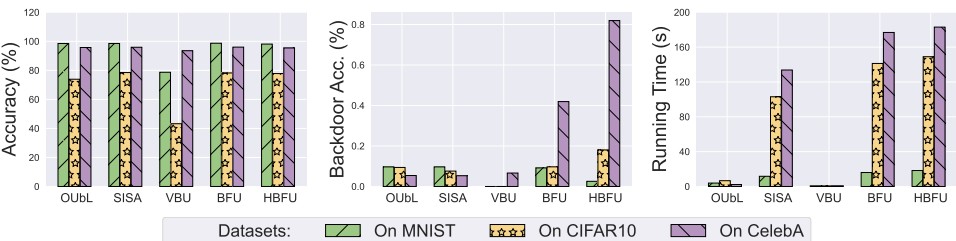

Figure 4: Evaluation of the effectiveness of different machine unlearning methods and datasets

achieves the best data removal effect (the lowest backdoor accuracy), it also compromises the model accuracy most, the lowest on all datasets. With the assistance of the other users to retrain the unlearned model, the federated unlearning methods (BFU and HBFU) achieve a similar model utility as OUbL and SISA. However, they may compromise data removal effects, worse than OUbL and SISA in most cases.

From the unlearning efficiency perspective, the third figure in Figure 4, OUbL achieves the second efficiency after the VBU. VBU is the most efficient because it implements unlearning solely based on the erased samples. Compared with centralized methods, the federated unlearning methods always consume more running time because FL-based methods require other normal users to participate in unlearning. Due to page limitation, additional experimental results are shown in Appendix F.

## 4.4 ABLATION STUDY OF CONSTRUCTED SAMPLES RATE AND AUXILIARY SAMPLES RATE

Clean Samples Rate (*CSR*) and Auxiliary Samples Rate (*ASR*) are two important parameters that influence the unlearning dataset construction of OUbL. In this section, we conduct experiments to study the influence of these two parameters. We also compare the privacy protection of OUbL with a version of OUbL that allows the server to know the unlearning intentions (abbreviated as "OUbL know unl. int.").

**Setup.** For ease of conducting experiments, we maintain a *CSR* : *ASR* ratio of 1 : 1 when constructing the uploading dataset. We set the $USR = 1\%$ on MNIST and CIFAR10 and $USR = 0.5\%$ on CelebA. We range *CSR* and *ASR* from 1% and 6% on MNIST and CIFAR10, and from 0.5% to 1% on CelebA. The privacy budget of VBU-LDP is set $\epsilon = 10$ here, and all the results are presented in Figure 5.

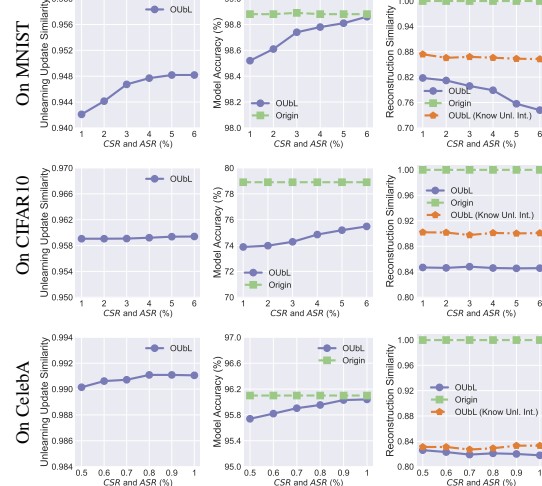

Figure 5: Evaluations for *CSR* and *ASR*

**Relationship between Unlearning Update Similarity and *CSR* and *ASR*.** Usually, with more clean samples and auxiliary samples, it would be easier to approach the unlearning update based on

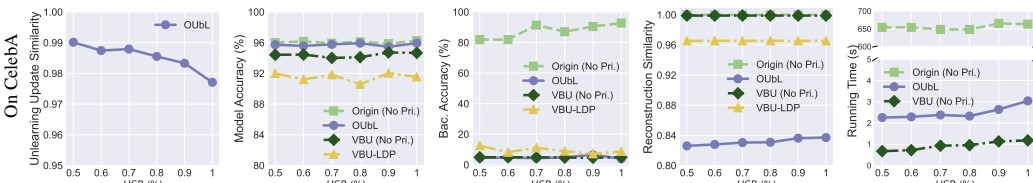

Figure 6: Evaluations of impact about different *USR*.

them. The first column in Figure 5 also demonstrates the trend. The unlearning update similarity increases as the *CSR* and *ASR* increase on all three datasets.

**Impact on Unlearning Effectiveness.** Since the data removal effect (backdoor accuracy) is stable with different *CSR* and *ASR*, we only demonstrate the model utility in Figure 5. It is obvious that the model accuracy increases when the *CSR* and *ASR* increases. The model utility degradation caused by the unlearning effect definitely can be mitigated if we have more clean and auxiliary data for continual learning updates for unlearning.

**Impact on Erased Data Protection.** The third column of Figure 5 demonstrates the privacy protection for the erased data against the reconstruction attacks. Higher *CSR* and *ASR* means more clean and auxiliary data is used to construct the uploading dataset. The unlearning noise is hidden in more samples, which increases the reconstruction difficulty, showing as lower reconstruction similarity in the third column in Figure 5. Compared with the "OUbL know unl. int.", the complete OUbL that hides the unlearning intentions significantly improves privacy protection, reflected by the huge decreased reconstruction similarity gap.

### 4.5 IMPACT OF UNLEARNING SAMPLES RATE (*USR*)

Figure 6 presents the evaluation of *USR* from 0.5% to 1% on CelebA. Additional evaluation of *USR* from 1% to 6% on MNIST and CIFAR10 is presented in Appendix F.2. *CSR* and *ASR* are set 0.5% for OUbL on CelebA. The privacy budget of VBU-LDP is set $\epsilon = 10$ here.

The first column of Figure 9 shows that more unlearning samples reduce the unlearning update similarity of OUbL when synthesizing the new dataset. However, OUbL still achieves stable unlearning effectiveness when *USR* increases, which is reflected by the model accuracy and backdoor accuracy. Although VBU achieves a thorough data removal effect, which is better than OUbL, we also note that the noise injection (VBU-LDP) will mitigate the data removal effect, as shown in the third figure on CelebA. Moreover, as shown in the last two figures in Figure 9, OUbL has slight privacy protection and efficiency decrease when *USR* increases.

## 5 SUMMARY AND FUTURE WORK

In this paper, we are the first to investigate privacy protection for both erased samples and unlearning intentions during machine unlearning. We propose an OUbL approach to solve this problem. OUbL constructs a new dataset for unlearning by incremental training, which hides the unlearning information in the constructed dataset to protect the erased data. Moreover, OUbL obliviously achieves the unlearning effect by incremental training on the constructed data using the original learning algorithm, hence not relying on customized unlearning algorithms and avoiding exposing the unlearning intentions to the servers. Our extensive experiments and comprehensive ablation studies have shown that the proposed OUbL can effectively protect the erased samples and the unlearning intentions while achieving a satisfactory unlearning effect. The proposed OUbL fulfills the gap between machine unlearning and privacy leakage to the server during unlearning, providing a powerful approach to implement machine unlearning with privacy protection.

As machine unlearning becomes increasingly important, our research serves as a stepping stone in understanding and protecting privacy during machine unlearning services. Future work should continue this line of inquiry, developing more privacy-preserving unlearning methods to uphold and support the right to be forgotten in MLaaS environments.

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

# A RELATED WORK

## A.1 MACHINE UNLEARNING

Machine unlearning techniques are motivated by the growing privacy concerns of individuals and the corresponding privacy regulations (Cao & Yang, 2015; Bourtoule et al., 2021). The most legitimate approach is retraining from scratch (Cao & Yang, 2015; Thudi et al., 2022). However, this method is often impractical due to the significant computational and storage costs involved, especially for complex deep-learning tasks. Consequently, numerous studies have sought to develop effective and efficient unlearning solutions (Yan et al., 2022; Warnecke et al., 2024).

Existing machine unlearning studies can be broadly categorized into *exact unlearning* (Cao & Yang, 2015; Bourtoule et al., 2021; Yan et al., 2022; Hu et al., 2024b) and *approximate unlearning* (Guo et al., 2020; Nguyen et al., 2020; Wang et al., 2024; Warnecke et al., 2024) methods. A representative exact unlearning method, introduced in (Bourtoule et al., 2021), extends naive retraining to reduce the computational cost of retraining a new model (Hu et al., 2024b; Yan et al., 2022). Exact unlearning completely removes the influence of the unlearned data on

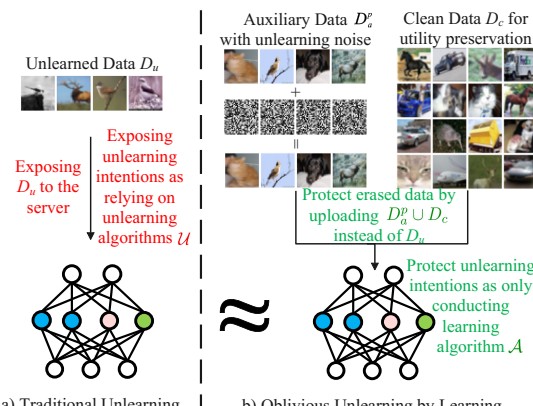

a) Traditional Unlearning | b) Oblivious Unlearning by Learning

Figure 7: Assuming a model with parameters $\theta$ trained using learning algorithm $\mathcal{A}$, traditional unlearning employs an unlearning algorithm $\mathcal{U}$ to unlearn $D_u$. Differently, OUbL constructs an updating dataset, comprising clean samples $D_c$ and unlearning noise-synthesized samples $D_a^p$, and uploads them to the server. The server only needs to incrementally update the model with the original learning algorithm $\mathcal{A}$ based on $D_c \cup D_a^p$, thereby guaranteeing the unlearning effect without uploading the erased data $D_u$ and without specific unlearning algorithms.

the model but requires significant storage space and is inefficient when removal requests are frequent. Conversely, approximate unlearning attempts to modify the model directly to approximate one retrained on the remaining dataset (Guo et al., 2020; Nguyen et al., 2020). Although more efficient than exact unlearning, approximate unlearning can lead to catastrophic unlearning (Nguyen et al., 2020; Wang et al., 2024; Nguyen et al., 2022).

## A.2 PRIVACY-PRESERVING MACHINE UNLEARNING

To facilitate addressing the challenging unlearning problem, all the aforementioned studies assume that the ML server is aware of users' data and unlearning intentions (Bourtoule et al., 2021; Cao & Yang, 2015; Guo et al., 2020; Sekhari et al., 2021). However, uploading the erased data and informing the server of unlearning intentions raises potential privacy threats to users, conflicting with the original intention of the "right to be forgotten" regulations (Wang et al., 2023; Liu et al., 2022b; Thudi et al., 2022). Although some studies highlight the privacy breaches caused by machine unlearning updates (Chen et al., 2021; Gao et al., 2022; Hu et al., 2024a; Zhang et al., 2023), only a few focus on privacy protection during machine unlearning, specifically avoiding the exposure of erased data and unlearning intentions to servers during unlearning. Most of these privacy-preserving machine unlearning solutions are based on federated learning, termed federated unlearning (Liu et al., 2022b; Wang et al., 2023; Su & Li, 2023; Gao et al., 2024).

Some federated unlearning methods (Wu et al., 2022; Fraboni et al., 2024; Liu et al., 2021; Lin et al., 2024) attempt to unlearn a user client's entire contribution from the trained FL model. They store all clients' uploaded parameters on the server side and estimate the unlearning user's influence based on these stored parameters (Fraboni et al., 2024; Liu et al., 2021). These approaches allow the FL server to implement unlearning without interacting with the user. However, it significantly degrades the original model's utility and is unsuitable for a user who wishes to unlearn only a small portion of their local dataset.

In contrast to unlearning a user's total contribution, the authors of (Liu et al., 2022b; 2021; Wang et al., 2023) investigated how to unlearn user-specified samples in FL. They proposed fast retraining methods based on the Hessian matrix and Bayesian inference (Box & Tiao, 2011). However, these approaches require the FL server to reactivate all users for retraining, which is impractical in real-world scenarios. Lin et al. (Lin et al., 2024) proposed a dynamic client selection method to avoid reactivating all user clients for unlearning. However, the vulnerability of privacy leakage from gradients remains. Gradients computed solely using the erased data for unlearning pose a higher risk of privacy leakage compared to standard FL training gradients, which are derived from the entire local dataset (Salem et al., 2020; Melis et al., 2019). Consequently, the FL framework can only offer limited protection for the privacy of erased data. Moreover, in existing unlearning methods (Bourtoule et al., 2021; Nguyen et al., 2020; Chundawat et al., 2023; Tarun et al., 2023), users must inform the server of their unlearning intentions so that the server can prepare customized operations for unlearning, which inevitably threatens the privacy of users' unlearning intentions.

Table 1: An overview of representative non-privacy protection and FL-based privacy-preserving unlearning methods.

| Machine Unlearning Methods | Service Scenarios | | Unlearning Data Size | | Unlearning Algorithms Necessity | | Unlearning Intention Protection | | Erased Data Protecting Methods |
|---|---|---|---|---|---|---|---|---|---|
| | Centralized Scenarios | Distributed Scenarios | Specified any samples | User's entire samples | Require unlearning algorithms | Not require unlearning algorithms | Need to inform servers | Not Need to inform servers | |
| SISA | ● | ○ | ● | ○ | ● | ○ | ● | ○ | No Protection |
| VBU | ● | ○ | ● | ○ | ● | ○ | ● | ○ | No Protection |
| BFU | ○ | ● | ● | ○ | ● | ○ | ● | ○ | By Gradients |
| HBFU | ○ | ● | ● | ○ | ● | ○ | ● | ○ | By Gradients |
| Federaser | ○ | ● | ○ | ● | ● | ○ | ● | ○ | By Gradients |
| OUbL (Ours) | ● | ○ | ● | ○ | ○ | ● | ○ | ● | Constructed new datasets |

●: the machine unlearning method is applicable; ○: the machine unlearning method is not applicable.
SISA (Bourtoule et al., 2021); VBU (Nguyen et al., 2020); BFU (Wang et al., 2023); HBFU (Liu et al., 2022b);Federaser (Liu et al., 2021).

### A.3 DIFFERENCE FROM EXISTING WORK

Our OUbL approach significantly differs from existing non-privacy protection machine unlearning methods (Bourtoule et al., 2021; Nguyen et al., 2020) and FL-based privacy-preserving unlearning methods (Liu et al., 2022b; 2021; Wang et al., 2023) in terms of service scenarios, unlearning data size, unlearning intentions protection, and erased data protection mechanisms, as depicted in Table 1. First, most existing privacy-preserving machine unlearning methods focus on federated (distributed) learning scenarios, utilizing gradients instead of original erased data to protect the privacy of erased samples (Liu et al., 2021; Wang et al., 2023; Liu et al., 2022b). Our method targets general centralized learning scenarios, where the centralized server processes model training, and most methods in these scenarios lack privacy protection (Bourtoule et al., 2021; Nguyen et al., 2020). Second, no studies currently protect unlearning intentions; all require informing the ML servers to customize unlearning algorithms. This work considers unlearning intentions as private information and aims to implement unlearning without needing specific unlearning algorithms, instead updating the model using the original learning algorithm.

We also note that some studies have attempted to generate adversarial or backdoor samples to influence model performance on targeted samples (Di et al., 2022; Madry et al., 2018; Zeng et al., 2023; Shafahi et al., 2019; Geiping et al., 2021). However, our approach differs in two significant ways. First, in backdooring methods (Geiping et al., 2021; Zeng et al., 2023), the targeted sample is typically a single instance, whereas unlearning scenarios usually involve multiple samples, increasing the complexity of noise generation. Second, the objective of backdooring is specific and relatively straightforward, i.e., altering the model's performance on a targeted sample. In contrast, the unlearning noise generation target, estimating the unlearning update, is considerably more challenging to determine.

## B EFFICIENT UNLEARNING UPDATE ESTIMATION ALGORITHM

We can use $\nabla_\theta^2 \ell((x_i, y_i); \theta_o)$ of any $(x_i, y_i)$ as an unbiased estimator of $H$. In Algorithm 2, we uniformly sample $t$ points $x_{s1}, x_{s2}, \ldots, x_{st}$ from a clean dataset $D_c$. Moreover, for a precise estimation, we repeat the process $r$ times and average the results to reduce variance. For each estimation round $k \in [1, 2, \ldots, r]$, we initially define $H_{k,0}^{-1} G_u = G_u$, where $G_u = \nabla_\theta \ell((x_u, y_u); \theta_o)$ for clarity, as shown in line 4 in Algorithm 2. Then, we recursively compute $H_{k,j}^{-1} G_u = G_u + (I -$

---

**Algorithm 2:** Efficient Unlearning Update Estimation

---

**Input:** Trained model $\theta_o$, the gradient vector of one erased sample $G_u = \nabla_\theta \ell((x_u, y_u); \theta_o)$, the clean samples used in model training $D_c = \{(x_i, y_i)\}_{i=1}^m$, Number of samples $t$, Number of iterations $r$
**Output:** Estimated unlearning update $H^{-1}\nabla_\theta \ell((x_u, y_u); \theta_o)$

1 **procedure** EUUE ($\theta_o$, $G_u$, $D_c$, $t$, $r$):
2     **for** $k \leftarrow 1$ **to** $r$ **do**
3        Sample $t$ points $\{x_{s_1}, x_{s_2}, ..., x_{s_t}\}$ uniformly from $D_c$
4        Initialize $H_{k,0}^{-1}G_u = G_u$
5        **for** $j \leftarrow 1$ **to** $t$ **do**
6           Compute $\nabla_\theta^2 L(x_{s_j}, \theta_o) H_{k,j-1}^{-1} G_u$;
7           Update $H_{k,j}^{-1}G_u = G_u + (I - \nabla_\theta^2 \ell((x_{s_j}, y_{s_j}); \theta_o)) H_{k,j-1}^{-1} G_u$;
8        Store result $H_{k,t}^{-1}G_u$
9     Average the results to get the final estimation: $H^{-1}G_u = \frac{1}{r}\sum_{k=1}^r H_{k,t}^{-1}G_u$
10     **return** $H^{-1}G_u$

---

$\nabla_\theta^2 \ell((x_{sj}, y_{sj}); \theta_o)) H_{k,j-1}^{-1}G_u$, as shown in lines 5 to 7 in Algorithm 2. We store the final round $H_{k,t}^{-1}G_u$ as the unbiased estimation of $H^{-1}G_u$ in the $k$-th iteration. After executing $r$ iterations, we average the results to obtain the final estimation.

## C    UNLEARNING NOISE DESCENT

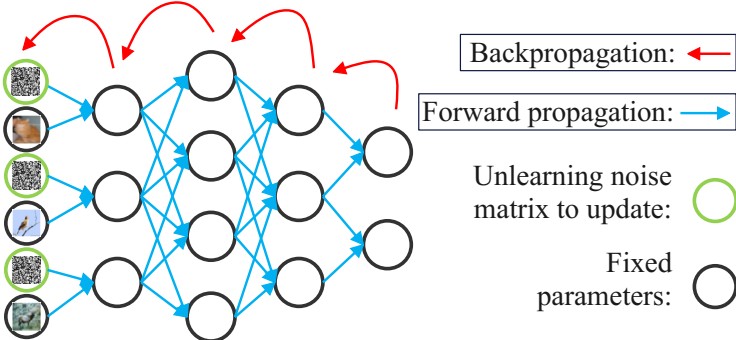

Figure 8: Unlearning noise descent. During the training process, we only update the unlearning noise matrixes (green circles) to find sufficient noise efficiently while fixing the model parameters (black circles).

Figure 8 shows the training process to find the suitable unlearning noise. Initially, we generate unlearning noise matrixes $\Delta^p$, shown as the green circles. During the training process, forward propagation, depicted by blue arrows, moves information from the input layer, which combines images with unlearning noise, through the network's hidden layers to the output layer. Concurrently, backpropagation, illustrated by red arrows, calculates the gradients of the loss according to Eq. (6) and adjusts the network's parameters by propagating errors backward. However, we here fix the network's parameters (black-bordered circles) and only update the unlearning noise matrixes (green circles) during the training process. After a few rounds of training, we can get sufficient unlearning noise for the auxiliary data to synthesize, ensuring the unlearning effect once the ML server incrementally trains the model based on the synthesized dataset.

The corresponding unlearning noise descent algorithm for the unlearning noise descent is presented in Algorithm 1.

# D    PROOF OF PROPOSITION 1

In this section, we provide the proof of Proposition 1 based on Zoutendijk, the Theorem 3.2 in (Nocedal & Wright, 2006). The main proof process is presented as follows.

*Proof.* Consider the incremental gradient descent update as the following form

$$\theta_o^{k+1} = \theta_o^k - \alpha_k \nabla \mathcal{L}_{inc}(\theta_o^k). \tag{12}$$

Due to Lipschitz smoothness of the gradient of the unlearning simulation loss $\mathcal{L}_{unl}$, we can estimate the value at $\theta_o^{k+1}$ by the descent lemma

$$\mathcal{L}_{unl}(\theta_o^{k+1}) \leq \mathcal{L}_{unl}(\theta_o^k) - \langle \alpha_k \nabla \mathcal{L}_{unl}(\theta_o^k), \nabla \mathcal{L}_{inc}(\theta_o^k) \rangle$$
$$+ \alpha_k^2 L \| \nabla \mathcal{L}_{inc}(\theta_o^k) \|^2, \tag{13}$$

where $L$ is the Lipschitz constant. If we use the cosine similarity identity:

$$\langle \nabla \mathcal{L}_{unl}(\theta_o^k), \nabla \mathcal{L}_{inc}(\theta_o^k) \rangle = \| \nabla \mathcal{L}_{inc}(\theta_o^k) \| \cdot \| \nabla \mathcal{L}_{unl}(\theta_o^k) \| \cos(\gamma^k), \tag{14}$$

where $\gamma^k$ denotes the angle between both gradients vectors, we can find that

$$\mathcal{L}_{unl}(\theta_o^{k+1}) \leq \mathcal{L}_{unl}(\theta_k^k) - \| \nabla \mathcal{L}_{inc}(\theta_o^k) \| \cdot \| \nabla \mathcal{L}_{unl}(\theta_o^k) \| \cos(\gamma^k)$$
$$+ \alpha_k^2 L \| \nabla \mathcal{L}_{inc}(\theta_o^k) \|^2 \tag{15}$$
$$= \mathcal{L}_{unl}(\theta_o^k) - (\alpha_k \frac{\| \nabla \mathcal{L}_{unl}(\theta_o^k) \|}{\| \nabla \mathcal{L}_{inc}(\theta_o^k) \|} \cos(\gamma^k) - \alpha_k^2 L) \| \nabla \mathcal{L}_{inc}(\theta_o^k) \|^2.$$

As such, the unlearning simulation loss decreases for nonzero step sizes if

$$\frac{\| \nabla \mathcal{L}_{unl}(\theta_o^k) \|}{\| \nabla \mathcal{L}_{inc}(\theta_o^k) \|} \cos(\gamma^k) > \alpha_k L, \tag{16}$$

i.e.,

$$\alpha_k L \leq \frac{\| \nabla \mathcal{L}_{unl}(\theta_o^k) \|}{\| \nabla \mathcal{L}_{inc}(\theta_o^k) \|} \frac{\cos(\gamma^k)}{c}, \tag{17}$$

for some $1 < c < \infty$. This follows from the assumption on the parameter $\beta$ in the statement of the Proposition 1. Reinserting this estimate into the descent inequality reveals that

$$\mathcal{L}_{unl}(\theta_o^{k+1}) < \mathcal{L}_{unl}(\theta_o^k) - \| \nabla \mathcal{L}_{unl} \|^2 \frac{\cos(\gamma^k)}{c'L}, \tag{18}$$

for $\frac{1}{c'} = \frac{1}{c} - \frac{1}{c^2}$. Due to monotonicity, we may sum over all descent inequalities, yielding

$$\mathcal{L}_{unl}(\theta_o^0) - \mathcal{L}_{unl}(\theta_o^{k+1}) \geq \frac{1}{c'L} \sum_{j=0}^{k} \| \nabla \mathcal{L}_{unl}(\theta_o^j) \|^2 \cos(\gamma^j). \tag{19}$$

As $\mathcal{L}_{unl}$ is bounded below, we may consider the limit of $k \to \infty$ to find

$$\sum_{j=0}^{\infty} \| \nabla \mathcal{L}_{unl}(\theta_o^j) \|^2 \cos(\gamma^j) < \infty. \tag{20}$$

If for all, except finitely many iterates the angle between adversarial and training gradient is less than $90°$, i.e., $\cos(\gamma^k)$ is bounded below by some fixed $\epsilon > 0$, as assumed, then the convergence to a stationary point follows:

$$\lim_{k \to \infty} \| \nabla \mathcal{L}_{unl}(\theta_o^k) \| \to 0. \tag{21}$$

$\square$

Table 2: Dataset statistics.

| Dataset | Feature Dimension | #. Classes | #. Samples |
|---|---|---|---|
| MNIST (Deng, 2012) | 28×28×1 | 10 | 70,000 |
| CIFAR10 (Krizhevsky et al., 2009) | 32×32×3 | 10 | 60,000 |
| CelebA (Liu et al., 2018) | 178×218×3 | 2 (Gender) | 202,599 |

# E  DETAILED EXPERIMENTAL SETTINGS

## E.1  DATASETS AND MODELS

**Datasets.** The statistics of all datasets used in our experiments are listed in Table 2. Both MNIST and CIFAR10 are used to train 10-class classification models. The experiment on CelebA is to identify the gender attributes of the face images. The task is a binary classification problem, different from the ones on MNIST and CIFAR10. We also introduce them as below.

- **MNIST.** MNIST contains 60,000 handwritten digit images for the training and 10,000 handwritten digit images for the testing. All these black and white digits are size normalized, and centered in a fixed-size image with $28 \times 28$ pixels.

- **CIFAR10.** CIFAR10 dataset consists of 60,000 32x32 colour images in 10 classes, with 6,000 images per class. There are 50,000 training images and 10,000 test images.

- **CelebA.** CelebA is a large-scale face attributes dataset with more than 200,000 celebrity images, each with 40 attribute annotations.

**Models.** We train a 5-layer MLP model on MNIST, a ResNet18 on CIFAR10 and a 7-layer CNN on CelebA. On the MNIST dataset, we set the learning rate $\eta = 0.001$. On CIFAR10 and CelebA, we set the learning rate $\eta = 0.0001$. During training, we set the minibatch size to 16 on MNIST and CIFAR10, and the minibatch size to 160 on CelebA. All algorithms are implemented using Pytorch 3.8 and are conducted on NVIDIA Quadro RTX 6000 GPUs.

## E.2  METRIC

- **Accuracy.** Model accuracy is calculated based on the test dataset, which shows if the unlearning methods influence the original ML service model utility.

- **Backdoor Accuary.** It calculated based on the backdoored dataset $D_{u,b}$ to evaluate the data removal effect.

- **Unlearning Update Similarity.** It calculates the cosine similarity between the estimated unlearning update by EUUE and the final OUbL unlearned update as

$$\text{sim}(\Delta\theta_{\textbf{EUUE}}, \Delta\theta_{\textbf{OUbL}}) = \frac{\Delta\theta_{\textbf{EUUE}} \cdot \Delta\theta_{\textbf{OUbL}}}{\|\Delta\theta_{\textbf{EUUE}}\| \cdot \|\Delta\theta_{\textbf{OUbL}}\|},$$

where $\Delta\theta_{\textbf{EUUE}}$ denotes the unlearning update estimated by EUUE (Algorithm 2) and $\Delta\theta_{\textbf{OUbL}}$ is the unlearning update after conducting OUbL. $\|\Delta\theta_{\textbf{EUUE}}\|$ and $\|\Delta\theta_{\textbf{OUbL}}\|$ are the Euclidean norms of the two updates vectors.

- **Reconstruction Similarity.** It calculates the cosine similarity between the unlearned samples $D_u$ and the attacks' reconstructed samples $\hat{D_u}$,

$$\text{sim}(D_u, \hat{D_u}) = \frac{D_u \cdot \hat{D_u}}{\|D_u\| \cdot \|\hat{D_u}\|}.$$

- **Running Time.** It is used to assess the efficiency, calculated by recording the time used in each training batch and multiplying it with the training epochs.

Table 3: Overall Evaluation Results on MNIST, CIFAR10 and CelebA.

| Single-Sample Unlearning | MNIST | | | | CIFAR10 | | | | CelebA | | | |
|---|---|---|---|---|---|---|---|---|---|---|---|---|
| | OUbL | BFU | SISA | VBU | OUbL | BFU | SISA | VBU | OUbL | BFU | SISA | VBU |
| Model Accuracy | 98.71% | **98.88%** | 98.75% | 97.91% | 76.75% | 77.61% | **79.37%** | 75.86% | 95.66% | 95.72% | **95.78%** | 95.80% |
| Backdoor Accuracy | 0.00% | 100% | 0.00% | 0.00% | 0.00% | 100% | 0.00% | 0.00% | 0.00% | 100% | 0.00% | 0.00% |
| Unl. Update Similarity | **0.976** | - | - | - | **0.989** | - | - | - | **0.998** | - | - | - |
| Reconstruction Similarity | **0.669** | 0.947 | 1 | 1 | **0.852** | 0.888 | 1 | 1 | **0.932** | 0.933 | 1 | 1 |
| Running time (s) | 4.178 | 15.95 | 11.87 | **0.024** | 8.231 | 122.26 | 113.00 | **0.088** | 0.432 | 168.61 | 135.42 | **0.033** |

| Multi-Sample Unlearning | MNIST, $USR = 1\%$ | | | | CIFAR10, $USR = 1\%$ | | | | CelebA, $USR = 0.5\%$ | | | |
|---|---|---|---|---|---|---|---|---|---|---|---|---|
| | OUbL | BFU | SISA | VBU | OUbL | BFU | SISA | VBU | OUbL | BFU | SISA | VBU |
| Model Accuracy | 98.52% | **98.70%** | 98.53% | 78.69% | 73.89% | 78.20% | **78.49%** | 43.23% | 95.74% | **96.01%** | 95.92% | 93.53% |
| Backdoor Accuracy | 9.67% | 9.16% | 9.67% | **0.00%** | 9.40% | 9.70% | 7.60% | **0.00%** | 4.55% | 41.92% | **4.41%** | 4.92% |
| Unl. Update Similarity | **0.942** | - | - | - | **0.959** | - | - | - | **0.992** | - | - | - |
| Reconstruction Similarity | **0.818** | 0.874 | 1 | 1 | **0.846** | 0.874 | 1 | 1 | **0.826** | 0.831 | 1 | 1 |
| Running time (s) | 3.920 | 16.03 | 11.70 | **0.631** | 6.633 | 141.26 | 103.01 | **0.587** | 2.257 | 176.86 | 133.74 | **0.672** |

BFU (Wang et al., 2023); SISA (Bourtoule et al., 2021); VBU (Nguyen et al., 2020)

### E.3 BENCHMARKS

- **SISA (Bourtoule et al., 2021).** The main process of SISA divides the full data $D$ into several shards $D^1, D^2, ..., D^k$ and trains sub-models with parameters $\theta^1, \theta^2, ..., \theta^k$ for each shard. When the server receives a request for unlearning sample $x_u$, it just needs to retrain the sub-model $\theta^i$ of shard $D^i$ that contains $x_u$. We set $k = 5$ disjoint shards and corresponding sub-models. We put the unlearned samples only on one shard, which is the ideal scenario of SISA.

- **VBU (Nguyen et al., 2020).** VBU is an approximate unlearning method based on variational Bayesian inference. For the convenience of experiments, we set a middle layer of original neural networks as the Bayesian layer and calculate the unlearning loss according to (Nguyen et al., 2020) based on the Bayesian layer and erased samples for unlearning.

- **BFU (Wang et al., 2023).** The BFU extended the variational Bayesian unlearning method to FL scenarios and proposed parameters self-sharing to mitigate the unlearning catastrophe. We implement BFU following the process as introduced in (Wang et al., 2023) and set the user number $k = 5$, and only one user needs unlearning, which is also an ideal scenario in FL.

- **HBFU (Liu et al., 2022b).** The HBFU extended the Hessian matrix-based unlearning methods (Sekhari et al., 2021; Guo et al., 2020) to FL scenarios. We implement HBFU following (Liu et al., 2022b) and also set the FL user number $k = 5$, and only one user needs unlearning.

## F ADDITIONAL EXPERIMENTS

### F.1 OVERALL EVALUATION OF OUbL

We first demonstrate the overall evaluation results of different machine unlearning methods on MNIST, CIFAR10 and CelebA, presented in Table 3. The bolded values indicate the best performance among the compared methods, while red-colored values signify results that are opposite from expectations. We fill a dash when the method does not contain the evaluation metrics.

**Setup.** We measure unlearning methods based on the five above-introduced evaluation metrics in single-sample and multi-sample unlearning scenarios. In the single samples unlearning scenario, only one unlearning sample needs to be unlearned. In the multi-sample unlearning scenario, we set the unlearning samples rate $USR = 1\%$, where $USR = \frac{m}{n}$, $m$ is the unlearned samples size and $n$ is the training data size. We illustrate the comparison of OUbL with one privacy-preserving federated unlearning method, BFU (Wang et al., 2023), one representative exact unlearning method, SISA (Bourtoule et al., 2021), and one approximate unlearning method, VBU (Nguyen et al., 2020).

**Evaluation of Unlearning Effectiveness.** The effect of machine unlearning is measured by model accuracy and backdoor accuracy, where the model accuracy represents the model utility after un-

learning, and the backdoor accuracy means if the backdoor-marked samples are unlearned from the model.

From the model utility perspective, in most cases, BFU or SISA achieves the best model accuracy. This is because SISA is a retraining-based method, and BFU needs all other normal federated users to assist in unlearning retraining to ensure the model's utility. Compared with the methods utilizing the remaining dataset, OUbL achieves a similar model accuracy as BFU and SISA, usually only slightly lower (not exceeding 2%) than them, much better than VBU, an approximate unlearning method without the assistance of the remaining dataset.

From the data removal perspective, OUbL and other centralized unlearning methods (SISA and VBU) can effectively unlearn the marked samples, reducing the backdoor accuracy to a very low level. For BFU, although the retraining assistance of other normal federated users helps to preserve model utility, it to some extent mitigates the unlearning update, failing to unlearn specified samples in the single-sample scenario on all datasets and in the multi-sample scenario on CelebA.

**Evaluation of Privacy Protection.** We evaluate privacy preservation through reconstruction similarity metrics. For OUbL and BFU, since the server cannot access the erased data, we conduct the reconstruction attacks (Hu et al., 2024a; Salem et al., 2020; Zhang et al., 2023) based on the unlearned update. Since the server of OUbL has no information about unlearning intentions, it treats the model updates as normal learning updates for attack (Salem et al., 2020).

The results of the three datasets show significant improvement in privacy protection by hiding unlearning intentions. For example, on MNIST, for SISA and VBU, since the server knows the unlearning requested samples, the reconstruction similarity is directly 1, the same as having no privacy preservation of these data. When gradient-based protection (BFU) is applied, the privacy protection of erased data achieves 0.947 reconstruction similarity when unlearning one sample. If we inform the server with unlearning intentions in OUbL, it achieves a reconstruction similarity to BFU. We present detailed comparisons for OUbL with informed unlearning intentions in Section 4.4. However, the server will be oblivious to unlearning when conducting OUbL, and the privacy protection of OUbL achieves 0.669 reconstruction similarity on single-sample unlearning on MNIST. OUbL achieves significant privacy protection on the three datasets compared with the FL-based and no-privacy protection unlearning methods. Although the FL-based method can also protect the privacy of erased data, it sometimes fails to unlearn samples and is inefficient as it needs the retraining assistance of other normal users.

**Evaluation of Efficiency.** Normally, OUbL achieves a slight efficiency improvement than BFU and SISA on MNIST and a significant improvement (more than $10\times$ speedup) on CIFAR10 and CelebA. Although BFU and SISA are much more efficient than naive retraining, they are still training time expensive compared with OUbL and VBU because BFU needs the retraining assistance of other users, and SISA needs to retrain $1/5$ shard of the original training dataset. VBU is the most efficient method, with less than 0.1 seconds for single sample unlearning, as it implements unlearning only based on the erased samples; however, the cost is the model utility degradation and lack of privacy protection.

F.2    ADDITIONAL EVALUATION OF IMPACT OF UNLEARNING SAMPLES RATE (*USR*)

**Setup.** Since the above unlearning methods have lots of similar metric values, for clear illustration and demonstration, we here only display the VBU and VBU-LDP methods as they implement machine unlearning based on solely erased samples, which requires the least data access. In this experiment, we range *USR* from 1% to 6% on MNIST and CIFAR10. *CSR* and *ASR* are set 1% for OUbL on MNIST and CIFAR10. The privacy budget of VBU-LDP is set $\epsilon = 10$ here. All the experimental results are presented in Figure 9.

**Relationship between Unlearning Update Similarity and *USR*.** The first column of Figure 9 shows the relationship between unlearning update similarity and *USR*. It is obvious that more unlearning samples in the unlearning request will increase the difficulty of constructing the dataset to achieve the unlearning effect, which is confirmed on the three datasets, higher *USR* decreasing the unlearning update similarity of OUbL a lot.

**Impact on Unlearning Effectiveness.** Unlearning effectiveness includes model utility preservation (the second column) and the data removal effect (the third column) in Figure 9.

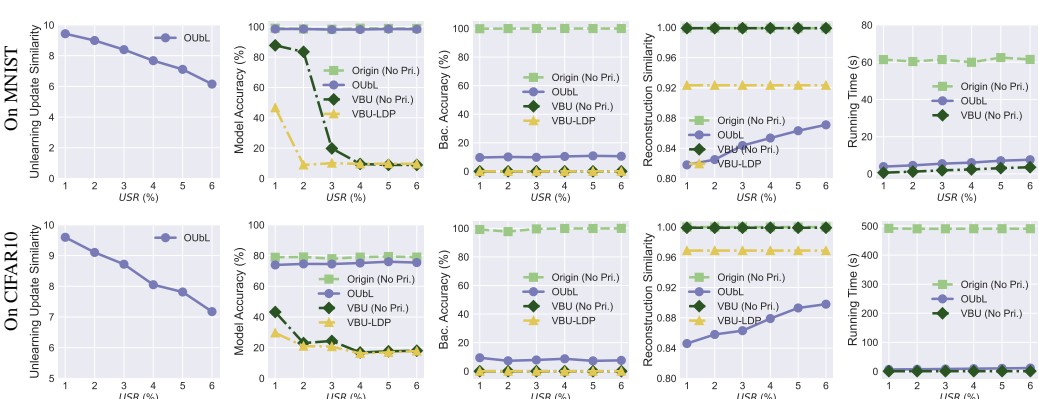

Figure 9: Evaluations of impact about different *USR*.

From the model utility preservation perspective, a good unlearning method should not be heavily influenced by the increasing *USR*. On all datasets, OUbL always achieves a similar model accuracy as the original trained model on all datasets even when *USR* increases from 1% to 6%. VBU only achieves an acceptable model accuracy on CelebA. Moreover, adding the LDP noise will definitely decrease the model utility, reflected by the huge gap between VBU and VBU-LDP.

From the data removal perspective, all the unlearning methods can effectively remove the influence of the marked backdooring samples from the model, reducing the backdoor accuracy of the original model.

**Impact on Erased Data Protection.** The fourth column in Figure 9 shows the privacy protection of the erased samples against the reconstruction attacks (Hu et al., 2024a; Zhang et al., 2023). For these methods with no privacy protection, the server directly has the erased sample, and the reconstruction similarity is 1. When *USR* increases, the reconstruction similarity of OUbL increases, too, meaning more privacy is leaked. Since we set $\epsilon = 10$ for VBU-LDP, OUbL always achieves better privacy protection than VBU-LDP.

**Impact on Unlearning Efficiency.** The fifth column in Figure 9 illustrates the running time of OUbL and VBU. The running time slightly increases as *USR* increases. Both OUbL and VBU achieve more than $20\times$ speedup than original training, which is much more efficient than most federated unlearning and retraining-based methods. Although OUbL consumes slightly more time than VBU, it is important to note that OUbL does not require uploading the erased data and informing the server for unlearning, which avoids privacy leakage of both erased data and unlearning intentions.

F.3 IMPACT OF UNLEARNING SAMPLES SIMILARITY

We know that the similarity between the erased samples and the remaining samples plays an important role in machine unlearning. In this section, we also conduct experiments to evaluate how the similarity influences the performance of OUbL.

**Setup.** To quantify the similarity between the erased samples and the remaining samples, we introduce a backdoor ratio $\beta$ to control the injected patch to the erased samples, formulated as $D_{u,\beta} \leftarrow (X_u + \beta \times patch, Y_u)$. We mixed both $D_{u,\beta}$ and original unlearned data $D_u$ into the training dataset for ML service model training. During the unlearning process, we only need to remove the contribution of the marked samples $D_{u,\beta}$. By adjusting $\beta$, we can simulate the different levels of similarity between the erased samples and the samples in the remaining dataset. This approach allows us to intuitively quantify the similarity between the unlearned samples and the remaining samples. The experimental results on MNIST, CIFAR10 and CelebA are presented in Figure 10.

**Relationship between Unlearned Data Similarity and Patch Injection Ratio.** The first column in Figure 10 illustrates the relationship between the patch injection ratio $\beta$ and similarity, where similarity is measured between the original samples $D_u$ and the patch-injected data $D_{u,\beta}$. In this experiment, we only need to unlearn $D_{u\beta}$. Across all datasets, including MNIST, CIFAR10, and CelebA, increasing the amount of noise (as indicated by a higher $\beta$) results in a decrease in simi-

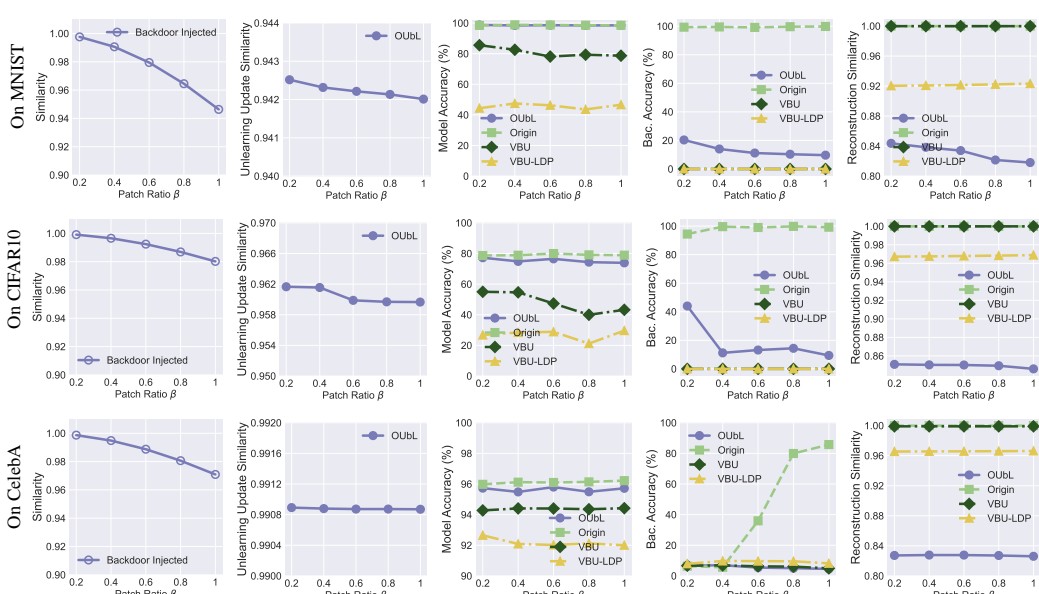

Figure 10: Evaluations for the impact of unlearning samples' similarity

larity. This trend is consistently observed, demonstrating that higher patch injection ratios make the unlearned samples more distinct from the original samples in the remaining dataset.

**Impact on Unlearning Update Similarity.** The second column in Figure 10 demonstrates the unlearning update similarity influenced by the patch ratio $\beta$. With more patches injected into the data, the marked samples will be more dissimilar to the original samples and the remaining samples. It increases the difficulty of OUbL in approaching the unlearning update via incremental learning, as the normal clean and the auxiliary data that we can use are not similar to the unlearned samples with a high patch injecting ratio. The unlearning update similarity obviously decreases as the patch ratio increases on all three datasets.

**Impact on Unlearning Effectiveness.** The third and fourth columns demonstrate the model utility and data removal of unlearning effectiveness. The data removal effect increases as the patch ratio increases, showing as the decreased backdoor accuracy. It means the more unique samples, larger $\beta$ and lower similarity, are easier to be unlearned. However, better data removal effect slightly decreases model utility, as showing from the model accuracy of all methods in the third column in Figure 10.

**Impact on Erased Data Protection.** Our OUbL achieves the best privacy protection for the erased samples, and the attacking difficulty for OUbL increases as the patch ratio increases. It is easy to understand because OUbL is more difficult to forge the unlearning update when patch ratio increases, which means the constructed data contains less unqiue information about the erased samples when $\beta$ is large. It makes the unlearning update less similar to the expected ones, and hence increase the reconstruction difficulty.

We omit the illustration of the unlearning efficiency results as the similarity not heavily influence the training time.

### F.4 Impact of Constructed Samples Rate and Auxiliary Samples Rate on Unlearning Efficiency

**Impact on Unlearning Efficiency.** As the synthesized dataset is constructed with the parameters *CSR* and *ASR*, the incremental training time based on the constructed dataset will definitely increase linearly as *CSR* and *ASR* increases, as shown in Figure 11.

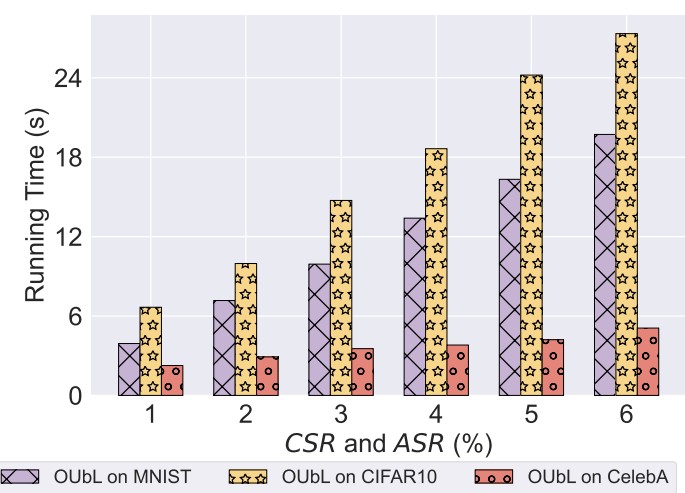

Figure 11: The impact of *CSR* and *ASR* about the running time. The *CSR* and *ASR* for CelebA is from 0.5% to 1%.

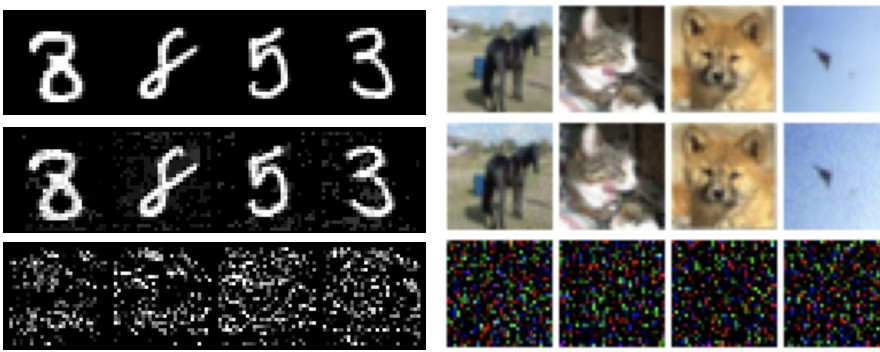

Figure R1: Example of the original auxiliary dataset (the first row), their noisy counterparts (the middle row), and directly construct data without an auxiliary dataset [R11] (the last row). @Reviewer iCGz

Table R1: Evaluating mixing unlearned data in the clean dataset on CIFAR10. The results demonstrate that mixing the unlearned samples into the constructed uploaded data for incremental learning negatively impacts the unlearning effect, as reflected by the increasing backdoor accuracy, but the model utility keeps. @Reviewer BZGC

| On CIFAR10 | Mixed 0% of Unlearned Data | 2% | 4% | 6% | 8% |
|---|---|---|---|---|---|
| Model Acc. | 73.89% | 73.85% | 73.78% | 73.25% | 73.03% |
| Backdoor Acc. | 9.40% | 13.60% | 33.40% | 35.40% | 43.26% |
| Running Time | 6.63 | 6.72 | 6.83 | 7.01 | 7.16 |

Table R2: Evaluating learning rate on MNIST and CIFAR10. The results demonstrate that a larger learning rate can speed the convergence to achieve unlearning, costing less computation and achieving a better unlearning effect (low backdoor accuracy by removing). The tradeoff is that it slightly decreases the model utility at the same time, which is not too much on MNIST but a little worse on CIFAR10. @Reviewer iCGz, @Reviewer Gp18

| | Metrics | Learning Rate: 0.0001 | 0.0002 | 0.0004 | 0.0006 | 0.0008 |
|---|---|---|---|---|---|---|
| | Model Acc. | 98.52% | 97.84% | 96.72% | 95.88% | 95.37% |
| On MNIST | Backdoor Acc. | 9.67% | 9.53% | 9.17% | 8.20% | 8.67% |
| | Running Time | 3.92 | 2.72 | 1.83 | 1.61 | 1.56 |
| | Model Acc. | 73.89% | 72.98% | 68.69% | 65.23% | 62.23% |
| On Cifar10 | Backdoor Acc. | 9.40% | 6.20% | 5.80% | 4.00% | 2.48% |
| | Running Time | 6.63 | 3.72 | 2.83 | 2.51 | 2.23 |

Table R3: Membership inference attack accuracy after unlearning by OUbL. The results demonstrate that OUbL can effectively reduce the MI accuracy, achieving a significant unlearning performance. @Reviewer 5Whi

| Dataset | Original Model | ASR and CSR, 1% | 2% | 3% | 4% |
|---|---|---|---|---|---|
| On MNIST | 63.86% | 53.87% | 53.61% | 53.02% | 52.86% |
| On CIFAR10 | 77.43% | 61.47% | 61.30% | 61.10% | 60.92% |
| On CelebA | 58.37% | 51.94% | 51.32% | 51.04% | 50.69% |

Table R4: The detailed running time. The results demonstrate that although we put more computational cost on the user side, it is affordable for users compared to the FL users in BFU. @Reviewer Ha5f

| Dataset | Total running time of OUbL (second) | Unlearning update estimation (User side) | Unlearning noise generation (User side) | Unlearning by incremental learning (Server side) | Total running time of BFU |
|---|---|---|---|---|---|
| On MNIST | 3.92 | 1.06 | 1.45 | 1.41 | 16.03 |
| On CIFAR10 | 6.64 | 1.02 | 2.10 | 3.52 | 141.26 |
| On CelebA | 2.26 | 0.72 | 0.83 | 0.71 | 176.86 |

Table R5: Experimental results on Adult. The task of the Adult dataset is to predict whether an individual's income exceeds \$50,000 per year, which is a binary classification. We first backdoor some samples in Adult by setting the "education-num" feature to 2 and changing the corresponding label. The aim of unlearning is to remove the influence of these backdoored samples, and the results are presented in the following table. Since the task on the Adult dataset is a binary task, dropping the backdoor accuracy of around 50% is similar to the random selection. Our method can effectively degrade the backdoor accuracy to around 50%, guaranteeing the effectiveness of unlearning. @Reviewer Gp18

| On Adult | Original | ASR and CSR, 1% | 2% | 3% | 4% |
|---|---|---|---|---|---|
| Model Acc. | 85.32% | 81.66% | 81.69% | 79.93% | 80.79% |
| Backdoor Acc. | 100.00% | 54.81% | 52.81% | 50.02% | 49.52% |
| Running Time | 15.31 | 0.54 | 1.03 | 1.51 | 1.93 |

# G    SCENARIOS AND THREAT MODEL

@Reviewer BZGC, @Reviewer iCGz, @Reviewer 5Whi, @Reviewer EoYb, @Reviewer Ha5f, @Reviewer Gp18

**Machine Unlearning Service Scenarios.** To facilitate understanding, we introduce the problem in a Machine Learning as a Service (MLaaS) scenario. In the MLaaS setting, there are two key entities: a ML server that trains models as ML services, and users (data owners) who contribute their data for ML model training. In such scenarios, machine unlearning occurs when users realize that some of their previously contributed samples are private and wish to revoke these data contributions from the trained models.

**The ML Server's Ability.** We assume the ML server is honest but curious [R1]: while it honestly hosts and provides ML services, including model training and updating, it may still be curious about private information, such as **unlearned data** and **unlearning intentions**, if there are other operations. Informing the server of unlearning intentions to customize unlearning operations is considered a privacy threat because it reveals users' unlearning purposes, potentially enabling the server to prepare targeted unlearning attacks [R1,R2]. Therefore, in our setting, we assume the ML server has only the learning algorithm $\mathcal{A}$ and the model with parameters $\theta$ to meet strict privacy requirements. The ML server will not conduct unlearning operations other than training the model using the learning algorithm $\mathcal{A}$ for model updating.

Moreover, we assume the ML server does not store the original training data and cannot access the erased data, which should not be exposed to the server again due to privacy concerns. This assumption is reasonable in both real-world and privacy-preserving MLaaS scenarios. In real-world applications, vast amounts of data are generated daily, leading to the need of prompt model updates. Consequently, many models are trained using incremental or continual learning techniques [R3,R4]. Therefore, the server does not retain the entire raw data due to its large size [R5,R6]. In privacy-preserving scenarios, the ML server is restricted from directly accessing private training data from users due to privacy concerns [R7,R8].

**The Users' Ability.** The training data $D$ was collected from all users and was used to train the model $\theta_o$. The unlearning user has the erased data $D_u \subset D$. To estimate the unlearning update as

the target for unlearning noise generation in our method, we assume the unlearning user can access the trained model $\theta_o$, which is a common setting even in many privacy-preserving scenarios such as FL. We assume the unlearning user has auxiliary clean examples $D_a$ so that they can synthesize a new dataset based on it with the unlearning noise, replacing the erased data $D_u$ for achieving the unlearning effect with only incremental learning using the synthesized dataset.

## H  DISCUSSION ABOUT DISTINGUISHING BENIGN UNLEARNING USERS AND MALICIOUS USERS @REVIEWER ICGZ

To distinguish a benign user who wants to delete their data from a malicious user and who wants to upload noisy gradients to sabotage the model performance, we can only propose some possible ways for the server to distinguish these two kinds of users. The most significant difference is the purposes of the unlearning user and the malicious user. Unlearning users want to remove some knowledge of their data from the model, and they also want to preserve the model's utility. Therefore, most clean samples and the auxiliary data they choose are in the same distribution as the genuine samples, and the synthesized noise should not influence the utility of the remaining dataset, as shown in the second objective of Eq.(4). However, the purpose of the malicious user is to sabotage the model performance. Their uploaded data will not be consistent with the genuine samples, so they can degrade model utility. We believe checking the similarity between the uploaded samples and genuine samples would be a possible solution. However, detailed poisoning attacking methods may need different solutions, and the problem is valuable to investigate in future work.

## I  DISCUSSION ABOUT DIFFERENCE BETWEEN EXISTING UNLEARNING METHODS @REVIEWER GP18

Compared with existing representative approximate unlearning methods [R19, R20, R21], our method also has the following differences. Specifically, the key techniques used in [R20] are the Hessian approximation and Fisher information, which is similar to our unlearning update estimation method that is also based on the Hessian matrix. The difference is that we use Hessian-vector products (HVPs) while [R20] uses the Fisher information to improve the efficiency. The HVPs solution is more efficient and more suitable to our scenarios in which the unlearning user cannot access the remaining dataset. [R19] and [R21] are approximate unlearning methods based on techniques called error maximizing. They generate error-maximizing noise for the unlearned samples to remove the influence from the model. One significant advantage of [R19] and [R21] is that they do not require access to the remaining training dataset. Compared with them, we put more effort into designing the method to further hide the unlearning data and the unlearning intentions from the server.

## J  ADDITIONAL EXPERIMENTS ON THE MORE PRACTICAL BLACK-BOX SCENARIOS @REVIEWER HA5F

To prove the feasibility of our method in the more practical black-box scenarios, we conducted additional experiments on the black-box setting on MNIST and CIFAR10. In this setting, the unlearning user cannot access the server's current model. The unlearning user only knows the type of the model (MLP on MNIST and CNN on CIFAR10 in our experiment), and the user only has the erasing data and the auxiliary data. We set the size of auxiliary data to 1% of the server-side training data. Other unlearning settings are the same as the main setting in the paper, where we first backdoor the erasing data for model training and aim to unlearn these backdoored erasing data.

With these settings, the unlearning user trains a shadow model ($\theta_s$) with 94.55% accuracy on MNIST and 42.57% accuracy on CIFAR10. By contrast, the accuracy of the server's model ($\theta_o$) trained with the entire dataset is 98.74% on MNIST and 78.80% on CIFAR10. Since both models are optimized on the erasing dataset, the proposed efficient unlearning update estimation (EUUE) method is effective for simulating the update of the unlearning data based on the shadow model. Hence, we can generate effective noise for the incremental learning data to approach the influence of unlearning. Then, we upload the constructed data to the server side for incremental learning, aiming to achieve the unlearning effect at the same time. We present the results as follows in Table R6.

Table R6: Additional experiments on the black-box setting. On both datasets, OUbL achieves effective unlearning performance, effectively removing the backdoor influence. The backdoor removal effectiveness in the black-box setting is slightly lower than in the white-box setting. However, the negative impact on the model utility is also mitigated. These experimental results demonstrate the feasibility of OUbL in a more practical scenario, which lets the unlearning user not rely on the assumption of white-box access to the trained model in the federated learning scenarios. @Reviewer Ha5f

| | Metrics | USR = 1% | 2% | 3% | 4% | 5% |
|---|---|---|---|---|---|---|
| On MNIST | Model Acc.(white-box) | 98.52% | 98.55% | 98.15% | 98.19% | 95.43% |
| | Model Acc. (black-box) | 98.26% | 98.20% | 98.31% | 98.27% | 98.54% |
| | Backdoor Acc. (white-box) | 9.67% | 10.08% | 9.83% | 10.42% | 10.57% |
| | Backdoor Acc. (black-box) | 12.33% | 9.58% | 11.67% | 10.64% | 11.83% |
| On Cifar10 | Model Acc.(white-box) | 73.89% | 74.57% | 74.50% | 75.15% | 75.99% |
| | Model Acc. (black-box) | 76.06% | 75.98% | 74.93% | 75.06% | 74.68% |
| | Backdoor Acc. (white-box) | 9.40% | 7.30% | 7.87% | 8.70% | 7.24% |
| | Backdoor Acc. (black-box) | 13.20% | 10.20% | 8.40% | 10.25% | 8.28% |

# K  ADDITIONAL REFERENCES

[R1]. Hu, Hongsheng, et al. "Learn what you want to unlearn: Unlearning inversion attacks against machine unlearning." IEEE Symposium on Security and Privacy (SP) (2024).

[R2]. Chen, Min, et al. "When machine unlearning jeopardizes privacy." Proceedings of the 2021 ACM SIGSAC conference on computer and communications security. 2021.

[R3]. Rolnick, David, et al. "Experience replay for continual learning." Advances in neural information processing systems 32 (2019).

[R4]. Lopez-Paz, David, and Marc'Aurelio Ranzato. "Gradient episodic memory for continual learning." Advances in neural information processing systems 30 (2017).

[R5]. Wu, Yue, et al. "Large scale incremental learning." Proceedings of the IEEE/CVF conference on computer vision and pattern recognition. 2019.

[R6]. Wang, Zifeng, et al. "Learning to prompt for continual learning." Proceedings of the IEEE/CVF conference on computer vision and pattern recognition. 2022.

[R7]. Naseri, Mohammad, Jamie Hayes, and Emiliano De Cristofaro. "Local and central differential privacy for robustness and privacy in federated learning." NDSS (2022).

[R8]. Bonawitz, Keith, et al. "Practical secure aggregation for privacy-preserving machine learning." proceedings of the 2017 ACM SIGSAC Conference on Computer and Communications Security. 2017.

[R9]. Gao, Xiangshan, et al. "Verifi: Towards verifiable federated unlearning." IEEE Transactions on Dependable and Secure Computing (2024).

[R10]. Guo, Xintong, et al. "Fast: Adopting federated unlearning to eliminating malicious terminals at server side." IEEE Transactions on Network Science and Engineering (2023).

[R11]. Zhu, Ligeng, Zhijian Liu, and Song Han. "Deep leakage from gradients." Advances in neural information processing systems 32 (2019).

[R12]. Kurmanji, M., Triantafillou, P., Hayes, J. and Triantafillou, E., 2024. Towards unbounded machine unlearning.Advances in neural information processing systems, 36.

[R13]. Chen, Min, et al. "When machine unlearning jeopardizes privacy." Proceedings of the 2021 ACM SIGSAC conference on computer and communications security. 2021.

[R14]. Hu, Hongsheng, et al. "Membership inference via backdooring." IJCAI (2022).

[R15]. Bertram, Theo, et al. "Five years of the right to be forgotten." Proceedings of the 2019 ACM SIGSAC Conference on Computer and Communications Security. 2019.

[R16]. Oh, Seong Joon, Bernt Schiele, and Mario Fritz. "Towards reverse-engineering black-box neural networks." Explainable AI: interpreting, explaining and visualizing deep learning (2019): 121-144.

[R17]. Salem, Ahmed, et al. "Updates-Leak: Data set inference and reconstruction attacks in online learning." 29th USENIX security symposium (USENIX Security 20). 2020.

[R18]. Wang, Binghui, and Neil Zhenqiang Gong. "Stealing hyperparameters in machine learning." 2018 IEEE symposium on security and privacy (SP). IEEE, 2018.

[R19] Chundawat, Vikram S., et al. "Zero-shot machine unlearning." IEEE Transactions on Information Forensics andSecurity 18 (2023): 2345-2354.

[R20] Golatkar, Aditya, Alessandro Achille, and Stefano Soatto. "Eternal sunshine of the spotless net: Selective forgettingin deep networks." Proceedings of the IEEE/CVF Conference on Computer Vision and Pattern Recognition. 2020.

[R21] Tarun, Ayush K., et al. "Fast yet effective machine unlearning." IEEE Transactions on Neural Networks and LearningSystems (2023).

