# OpenReview forum: "Oblivious Unlearning by Learning: Machine Unlearning Without Exposing Erased Data"
_ICLR.cc/2025/Conference — Submitted to ICLR 2025_

### Official Review · Reviewer_Gp18 · 2024-10-29

**Soundness:** 2
**Presentation:** 3
**Contribution:** 3
**Rating:** 6
**Confidence:** 5

**Summary:**

This paper focuses on the machine unlearning problem. The motivation for this paper is the observation that most existing works require the user to upload their original data for unlearning, which might disobey the purpose of machine unlearning. Thus, the authors propose a new machine unlearning method by learning. The model owner updates the target model on synthetic unlearning data, and the user does not need to request for unlearning. Extensive experiments validate the effectiveness of the proposed method.

**Strengths:**

1, This paper is well-written and easy to understand.

2, A new unlearning method is proposed, and it solves the existing method’s limitations of uploading the unlearned data to the server.

3, Extensive experiments on several datasets and model architectures validate the effectiveness of the proposed method.

**Weaknesses:**

1, Some important justifications are missing from the attack model. First, the idea of the proposed attack is motivated by the claim that users do not want to leak the unlearning intention because of potential unlearning attacks. Then, unlearning is hidden by the server updating the model. In this case, my question is why the server would update the model as the user wishes. I mean, the server decides the update itself. I did not see how this works. Second, even if the server implements the update, what if the server filters out unlearning samples?  From the current version, it seems the authors miss justifications on the two points.

2, The proposed method’s applicability is in question. For example, how does the learning rate affect the effectiveness of this method? If the server keeps the learning rate as a secret, can the proposed method work?

3, There are several existing works discussing unlearning without seeing the original unlearning data. For example, the following papers. The authors should have a clear discussion on them.

[R1] Chundawat, Vikram S., et al. "Zero-shot machine unlearning." IEEE Transactions on Information Forensics and Security 18 (2023): 2345-2354.

[R2] Golatkar, Aditya, Alessandro Achille, and Stefano Soatto. "Eternal sunshine of the spotless net: Selective forgetting in deep networks." Proceedings of the IEEE/CVF Conference on Computer Vision and Pattern Recognition. 2020.

[R3] Tarun, Ayush K., et al. "Fast yet effective machine unlearning." IEEE Transactions on Neural Networks and Learning Systems (2023).

4, The proposed method does not use the forget data but seems to use retain data. Is this reasonable? Why the server can access and retain data? If the server can access the retained data, then it mean the server kept the original training data? If so, it seems to violate the original assumption.

5, The experiments are conducted on only image datasets. Can the proposed method be used for other data types? For example, text or tabular data.

**Questions:**

My main questions are with my concerns raised in the weakness part.

1. Can you provide justifications in regard to weakness one?

2, Can you provide an answer to the applicability of this method in weakness two?

3, How does the proposed method differ from existing works?

4, Can the server easily defend against the proposed unlearning method?

5, Can the proposed method work for other data types?

---

> ### Author Response · Authors · 2024-11-20
> **Rebuttal by Authors**
>
> # Rebuttal Part 1
>
>
> We sincerely appreciate Reviewer Gp18 for providing a detailed summary of our strengths. We also greatly thank Reviewer Gp18 for proposing these insightful questions. We uploaded supplementary materials, which separate the newly added context for your convenience. Below, we provide our responses to the comments, denoted by [W] for weaknesses and [Q] for questions.
>
>
> **Response to W1 and Q1:**
> We greatly appreciate the Reviewer's insightful comment. In response, we have enhanced our manuscript by adding a detailed discussion of the scenario and threat model. Specifically, we frame our approach within the context of a continual learning scenario, where incremental learning is performed frequently and systematically. This setting is particularly relevant and popular, as users continuously generate vast amounts of new data daily, leading to the need for timely model updates.
> Additionally, we emphasize that our method is designed to prevent the server from identifying the unlearned data or detecting unlearning intentions. To achieve this, the server only receives the updated data containing unlearning noise for model updates, making it challenging for the server to reconstruct the original unlearning samples.
> We present the detailed scenario and threat model as follows.
>
> >**Machine Unlearning Service Scenarios.** To facilitate understanding, we introduce the problem in a Machine Learning as a Service (MLaaS) scenario. In the MLaaS setting, there are two key entities: a ML server that trains models as ML services, and users (data owners) who contribute their data for ML model training. In such scenarios, machine unlearning occurs when users realize that some of their previously contributed samples are private and wish to revoke these data contributions from the trained models.
>
> >**The ML Server's Ability.** We assume the ML server is honest but curious [R1]: while it honestly hosts and provides ML services, including model training and updating, it may still be curious about private information, such as **unlearned data** and **unlearning intentions**, if there are other operations. Informing the server of unlearning intentions to customize unlearning operations is considered a privacy threat because it reveals users' unlearning purposes, potentially enabling the server to prepare targeted unlearning attacks [R1,R2]. Therefore, in our setting, we assume the ML server has only the learning algorithm $\mathcal{A}$ and the model with parameters $\theta$ to meet strict privacy requirements. The ML server will not conduct unlearning operations other than training the model using the learning algorithm $\mathcal{A}$ for model updating.
>
> >Moreover, we assume the ML server does not store the original training data and cannot access the erased data, which should not be exposed to the server again due to privacy concerns. This assumption is reasonable in both real-world and privacy-preserving MLaaS scenarios. In real-world applications, vast amounts of data are generated daily, leading to the need for prompt model updates. Consequently, many models are trained using incremental or continual learning techniques [R3,R4]. Therefore, the server does not retain the entire raw data due to its large size [R5,R6]. In privacy-preserving scenarios, the ML server is restricted from directly accessing private training data from users due to privacy concerns [R7,R8].
>
> >**The Users' Ability.** The training data $D$ was collected from all users and was used to train the model $\theta_o$. The unlearning user has the erased data $D_u \subset D$. To estimate the unlearning update as the target for unlearning noise generation in our method, we assume the unlearning user can access the trained model $\theta_o$, which is a common setting even in many privacy-preserving scenarios such as FL. We assume the unlearning user has auxiliary clean examples $D_a$ so that they can synthesize a new dataset based on it with the unlearning noise, replacing the erased data $D_u$ for achieving the unlearning effect with only incremental learning using the synthesized dataset.
>
> References [R1-R8] are provided at the end of the revised manuscript and the first comment.

---

> > ### Author Response · Authors · 2024-11-20
> > **Rebuttal Part 2**
> >
> > # Rebuttal Part 2
> >
> > **Response to W2 and Q2:**
> > We sincerely appreciate the Reviewers' questions and comments. In the above-provided scenario, our method is applicable and feasible, which can be supported by the results in the paper. To answer the question of whether our method is applicable to different learning rates, we conduct additional experiments, and the results are presented in the following table. The results demonstrate that a larger learning rate can speed the convergence to achieve unlearning, costing less computation and achieving a better unlearning effect (low backdoor accuracy by removing). The tradeoff is that it slightly decreases the model utility at the same time, which is not too much on MNIST but a little worse on CIFAR10.
> >
> > The **Table R2** of evaluating learning rate on MNIST and CIFAR10: USR=1%, ASR=1%, and CSR=1%:
> >
> > | On MNIST       | Learning Rate: 0.0001|  0.0002  | 0.0004  |  0.0006 | 0.0008 |
> > | --------         | --------    | -------- | -------- |  -------- |   -------- |
> > | Model Acc.       | 98.52%      | 97.84%   |  96.72%  |  95.88%   |  95.37%    |
> > | Backdoor Acc.    | 9.67%       | 9.53%   | 9.17%   |  8.20%   |  8.67%     |
> > | Running Time     | 3.92        | 2.72     | 1.83     |  1.61     |  1.56      |
> >
> >
> > | On CIFAR10       | Learning Rate: 0.0001|  0.0002  | 0.0004  |  0.0006 |    0.0008 |
> > | --------         | --------    | -------- | -------- |  -------- |     --------    |
> > | Model Acc.       | 73.89%      | 72.98%   |  68.69%  |   65.23%  |     62.23%      |
> > | Backdoor Acc.    | 9.40%       | 6.20%   |  5.80%    |    4.00%  |    2.48%        |
> > | Running Time     | 6.63        | 3.72     | 2.83     |  2.51     |    2.23        |
> >
> >
> >
> >
> >
> >
> > **Response to W3 and Q3:**
> > We greatly appreciate the Reviewers' questions and comments. We add a new discussion about the difference between our method and existing works. Specifically, the key techniques used in [R20] are the Hessian approximation and Fisher information, which is similar to our unlearning update estimation method that is also based on the Hessian matrix. The difference is that we use Hessian-vector products (HVPs) while [R20] uses the Fisher information to improve the efficiency. The HVPs solution is more efficient and more suitable to our scenarios in which the unlearning user cannot access the remaining dataset. [R19] and [R21] are approximate unlearning methods based on techniques called error maximizing. They generate error-maximizing noise for the unlearned samples to remove the influence from the model. One significant advantage of [R19] and [R21] is that they do not require access to the remaining training dataset. Compared with them, we put more effort into designing the method to further hide the unlearning data and the unlearning intentions from the server. We present the revised difference discussion in the Appendix and show it as follows.
> >
> > >Compared with existing representative approximate unlearning methods [R19, R20, R21], our method also has the following differences. Specifically, the key techniques used in [R20] are the Hessian approximation and Fisher information, which is similar to our unlearning update estimation method that is also based on the Hessian matrix. The difference is that we use Hessian-vector products (HVPs) while [R20] uses the Fisher information to improve the efficiency. The HVPs solution is more efficient and more suitable to our scenarios in which the unlearning user cannot access the remaining dataset. [R19] and [R21] are approximate unlearning methods based on techniques called error maximizing. They generate error-maximizing noise for the unlearned samples to remove the influence from the model. One significant advantage of [R19] and [R21] is that they do not require access to the remaining training dataset. Compared with them, we put more effort into designing the method to further hide the unlearning data and the unlearning intentions from the server.
> >
> > [R19] Chundawat, Vikram S., et al. "Zero-shot machine unlearning." IEEE Transactions on Information Forensics andSecurity 18 (2023): 2345-2354.
> >
> > [R20] Golatkar, Aditya, Alessandro Achille, and Stefano Soatto. "Eternal sunshine of the spotless net: Selective forgettingin deep networks." Proceedings of the IEEE/CVF Conference on Computer Vision and Pattern Recognition. 2020.
> >
> > [R21] Tarun, Ayush K., et al. "Fast yet effective machine unlearning." IEEE Transactions on Neural Networks and LearningSystems (2023).

---

> > > ### Author Response · Authors · 2024-11-20
> > > **Rebuttal Part 3**
> > >
> > > # Rebuttal Part 3
> > >
> > > **Response to W4 and Q4:**
> > > We sincerely thank the Reviewer's comments and questions. Our method does not require access to the remaining data, which is supplemented in the discussion of the above-presented scenarios. Instead, in our method, the unlearning users require the original model, the unlearned data, and some new auxiliary data for incremental learning. The server does not need to store any data. The server only receives the unlearning uploaded samples and conducts incremental learning based on the received data.
> > >
> > > Additionally, to respond to the **Q4**, we should emphasize that the server is hard to against the proposed unlearning method. Our method is designed to prevent the server from identifying the unlearned data or detecting unlearning intentions. To achieve this, the server only receives the updated data containing unlearning noise for model updates. It is challenging for the server to defend against the unlearning or infer the information about the unlearning as the server does not even know the unlearning intentions.
> > >
> > >
> > >
> > > **Response to W5 and Q5:**
> > > We greatly thank the Reviewer's questions and suggestions. The proposed method is easily extended to other types of data. We additionally conducted the experiment on the tabular dataset, Adult. The corresponding results are presented in the following table.
> > >
> > > The task of the Adult dataset is to predict whether an individual's income exceeds $50,000 per year, which is a binary classification. We first backdoor some samples in Adult by setting the "education-num" feature to 2 and changing the corresponding label. The aim of unlearning is to remove the influence of these backdoored samples, and the results are presented in the following table. Since the task on the Adult dataset is a binary task, dropping the backdoor accuracy of around 50% is similar to the random selection. Our method can effectively degrade the backdoor accuracy to around 50%, guaranteeing the effectiveness of unlearning.
> > >
> > > The **Table R5** of experimental results on Adult:
> > >
> > > | On Adult         | Original    |  ASR and CSR, 1% | 2% |  3% |    4% |
> > > | --------         | --------    | --------  | -------- |  -------- |     --------    |
> > > | Model Acc.       | 85.32%      | 81.66%   |  81.69%  |  79.93%  |     80.79%      |
> > > | Backdoor Acc.    | 100.00%     | 54.81%   |  52.81%  |  50.02% |    49.52%        |
> > > | Running Time     | 15.31       | 0.54     | 1.03    |  1.51     |    1.93        |

---

> > > > ### Author Response · Authors · 2024-11-25
> > > > **Reminder on follow-up discussion (one day left before rebuttal ends)**
> > > >
> > > > Dear Reviewer Gp18,
> > > >
> > > > We sincerely appreciate the time and effort you've invested in reviewing our manuscript. Your expertise and dedication to the review process are truly invaluable to us.
> > > >
> > > > As the discussion phase draws to a close in one day, we kindly request you to provide your feedback on our rebuttal. Your insights are of immense importance to us, and we look forward to any additional comments you may have. If our replies have addressed your concerns, we would be thankful for your recognition of this. Should there be any points requiring further discussion or explanation, please feel free to contact us. Furthermore, we are fully prepared to engage more to enhance our work during this pivotal stage.
> > > >
> > > > Best regards,
> > > >
> > > > Authors

---

> > > > > ### Comment · Reviewer_Gp18 · 2024-11-25
> > > > > **Thanks for the responses.**
> > > > >
> > > > > I am in general happy with the response. I have raised my scores.

---

> > > > > > ### Author Response · Authors · 2024-11-25
> > > > > > **Thank you for your insights and raising the score**
> > > > > >
> > > > > > Dear Reviewer Gp18,
> > > > > >
> > > > > > We sincerely thank you for your insightful comments with us and are pleased to hear that our responses are generally satisfactory. We thank you again for elevating the quality of our work.
> > > > > >
> > > > > > Best regards,
> > > > > >
> > > > > > Authors

---

### Official Review · Reviewer_Ha5f · 2024-11-02

**Soundness:** 2
**Presentation:** 3
**Contribution:** 2
**Rating:** 3
**Confidence:** 3

**Summary:**

The paper proposes an Oblivious Unlearning by Learning (OUbL) approach to address the challenge of implementing unlearning without revealing erased data or unlearning intentions to the server. The approach involves constructing a new dataset with synthesized unlearning noise to achieve this goal. While the idea is novel and inspiring, there are several concerns regarding the threat model, practicality, and computational burden on the user that require further clarification.

**Strengths:**

- The concept of using synthesized unlearning noise is innovative and offers a new perspective on achieving oblivious unlearning.
- The paper addresses a timely and important problem in the field of machine learning, with implications for privacy-preserving model updates.
- The authors’ efforts in designing the mechanism to generate unlearning noise demonstrate creativity and technical insight.

**Weaknesses:**

- Lack of a clear threat model: The manuscript does not clearly describe the threat model, making it difficult to assess the security and assumptions under which the proposed approach operates. Specifically, there is no detailed explanation of the knowledge and capabilities a user must have to estimate unlearning updates and construct the synthesized auxiliary dataset.

- Practicality concerns: The proposed method appears to require white-box access to the model, particularly for the noise synthesis step through gradient matching. This may limit the real-world applicability of the approach, as most users may not have such access or capability in common unlearning scenarios.

- Computational burden on the user: The approach seems to shift a significant amount of computational cost from the server to the user, particularly in generating unlearning noise. The authors do not provide an analysis of the computational cost for users, leaving unanswered whether it is feasible for normal users to request unlearning in practice.

**Questions:**

1. Can the authors provide a clearer description of the threat model, particularly defining the knowledge and capabilities that a user needs to perform unlearning?

2. In what real-world scenarios can a normal user have white-box knowledge of the model, and how can the proposed approach be applied in such settings? Could the authors discuss practical situations where this would be feasible?

3. How much computational cost is incurred by users when requesting unlearning, and is this burden affordable for typical users? Would it be possible to provide quantitative analysis or benchmarks to showcase the overhead for users?

---

> ### Author Response · Authors · 2024-11-20
> **Rebuttal by Authors**
>
> # Rebuttal Part 1
>
>
> We sincerely appreciate Reviewer Ha5f for providing a detailed summary of our strengths. We also greatly thank Reviewer Ha5f for proposing these insightful questions. We worked diligently to address all the concerns and supplemented the discussion about the threat model, which is provided in the revised paper. We also uploaded supplementary materials, which separate the newly added context for your convenience. Below, we provide our responses to the comments, denoted by [W] for weaknesses and [Q] for questions.
>
>
> **Response to W1 and Q1:** We greatly appreciate the Reviewer's insightful comment. To address the concern raised, in our method, the estimation of the unlearning influence for synthesizing the auxiliary data requires access to the trained original model, denoted as $\theta_o$, unlearned samples $D_u$, and some auxiliary clean examples $D_a$. We have supplemented our discussion on the threat model as follows.
>
> >**Machine Unlearning Service Scenarios.** To facilitate understanding, we introduce the problem in a Machine Learning as a Service (MLaaS) scenario. In the MLaaS setting, there are two key entities: a ML server that trains models as ML services, and users (data owners) who contribute their data for ML model training. In such scenarios, machine unlearning occurs when users realize that some of their previously contributed samples are private and wish to revoke these data contributions from the trained models.
>
> >**The ML Server's Ability.** We assume the ML server is honest but curious [R1]: while it honestly hosts and provides ML services, including model training and updating, it may still be curious about private information, such as **unlearned data** and **unlearning intentions**, if there are other operations. Informing the server of unlearning intentions to customize unlearning operations is considered a privacy threat because it reveals users' unlearning purposes, potentially enabling the server to prepare targeted unlearning attacks [R1,R2]. Therefore, in our setting, we assume the ML server has only the learning algorithm $\mathcal{A}$ and the model with parameters $\theta$ to meet strict privacy requirements. The ML server will not conduct unlearning operations other than training the model using the learning algorithm $\mathcal{A}$ for model updating.
>
> >Moreover, we assume the ML server does not store the original training data and cannot access the erased data, which should not be exposed to the server again due to privacy concerns. This assumption is reasonable in both real-world and privacy-preserving MLaaS scenarios. In real-world applications, vast amounts of data are generated daily, leading to the need for prompt model updates. Consequently, many models are trained using incremental or continual learning techniques [R3,R4]. Therefore, the server does not retain the entire raw data due to its large size [R5,R6]. In privacy-preserving scenarios, the ML server is restricted from directly accessing private training data from users due to privacy concerns [R7,R8].
>
> >**The Users' Ability.** The training data $D$ was collected from all users and was used to train the model $\theta_o$. The unlearning user has the erased data $D_u \subset D$. To estimate the unlearning update as the target for unlearning noise generation in our method, we assume the unlearning user can access the trained model $\theta_o$, which is a common setting even in many privacy-preserving scenarios such as FL. We assume the unlearning user has auxiliary clean examples $D_a$ so that they can synthesize a new dataset based on it with the unlearning noise, replacing the erased data $D_u$ for achieving the unlearning effect with only incremental learning using the synthesized dataset.
>
> References [R1-R8] are provided at the end of the revised manuscript and the first comment.

---

> > ### Author Response · Authors · 2024-11-20
> > **Rebuttal Part 2**
> >
> > # Rebuttal Part 2
> >
> > **Response to W2 and Q2:** We greatly thank the Reviewer's insightful comment. We do require the trained model $\theta_o$ to estimate the unlearning influence and conduct the gradient matching. We have supplemented the discussion about the rationality of the assumption in the above threat model. In many scenarios, even privacy-preserving scenarios such as FL, the users (as the data contributor) can access the trained models. Our method can be directly applied in these scenarios. However, as noted by the Reviewer, in a more practical scenario where unlearning users do not have access to the trained models, we acknowledge the challenge and propose potential solutions to construct shadow models. Unlearning users may employ hyperparameter stealing attacks to construct shadow models, a technique demonstrated to effectively replicate server models, as shown in [R16, R17, R18].
> > Subsequently, these users can apply our proposed method to synthesize unlearning noise. This is a potential solution we can think of at this point. We believe this question is great and valuable to investigate in future work.
> >
> >
> > [R16]. Oh, Seong Joon, Bernt Schiele, and Mario Fritz. "Towards reverse-engineering black-box neural networks." Explainable AI: interpreting, explaining and visualizing deep learning (2019): 121-144.
> >
> > [R17]. Salem, Ahmed, et al. "{Updates-Leak}: Data set inference and reconstruction attacks in online learning." 29th USENIX security symposium (USENIX Security 20). 2020.
> >
> > [R18]. Wang, Binghui, and Neil Zhenqiang Gong. "Stealing hyperparameters in machine learning." 2018 IEEE symposium on security and privacy (SP). IEEE, 2018.
> >
> >
> >
> >
> > **Response to W3 and Q3:** We greatly appreciate the Reviewer's insightful comment. In the paper, the computational cost is the entire cost for unlearning, including the server and the unlearning user, which shows competitiveness compared with most existing unlearning methods, as our method only involves the unlearned dataset $D_u$ and an auxiliary dataset $D_a$ with the same size as the unlearned dataset. As you mentioned, our method shifts partial computational costs from the server to the user. We supplemented the detailed computational cost in the table below, which divides the computational costs between the unlearning user and the server. These experiments are conducted with the unlearning samples rate (USR) of 1%. Clean samples rate (CSR) and auxiliary samples rate (ASR) are also 1%. The results demonstrate that although we put more computational cost on the user side, it is affordable for users compared to the FL users in BFU. This is because our designed method only relies on the unlearned dataset, which is significantly smaller than the remaining dataset and the remaining local dataset of the user. Hence, the computational cost of the unlearning user in our method is smaller than the methods that require users to assist unlearning with remaining local data.
> >
> >
> > The **Table R4** running time on MNIST, CIFAR10, and CelebA:
> >
> > | Dataset   | Total running time of OUbL (Second) | Unlearning update estimation (User side)    | Unlearning noise generation (User side)     |  Unlearning by incremental learning (Server side) | Total running time of BFU |
> > | --------  | --------    | -------- | -------- |  -------- |   -------- |
> > | MNIST     | 3.92        | 1.06   |  1.45      |  1.41     |  16.03     |
> > | CIFAR10   | 6.64        | 1.02    | 2.10      |  3.52     |  141.26    |
> > | CelebA    | 2.26        | 0.72    | 0.83      |  0.71     |  176.86    |

---

> > > ### Author Response · Authors · 2024-11-25
> > > **Reminder on follow-up discussion (one day left before rebuttal ends)**
> > >
> > > Dear Reviewer Ha5f,
> > >
> > > We sincerely appreciate the time and effort you've invested in reviewing our manuscript. Your expertise and dedication to the review process are truly invaluable to us.
> > >
> > > As the discussion phase draws to a close in one day, we kindly request you to provide your feedback on our rebuttal. Your insights are of immense importance to us, and we look forward to any additional comments you may have. If our replies have addressed your concerns, we would be thankful for your recognition of this. Should there be any points requiring further discussion or explanation, please feel free to contact us. Furthermore, we are fully prepared to engage more to enhance our work during this pivotal stage.
> > >
> > > Best regards,
> > >
> > > Authors

---

> > > > ### Comment · Reviewer_Ha5f · 2024-11-25
> > > >
> > > > Thank you for your response.
> > > >
> > > > However, my primary concerns remain unresolved. A significant issue lies in the required user capabilities. The proposed approach assumes that users have white-box access to the model, which may not be practical in many real-world applications. Furthermore, the additional requirement of constructing shadow models raises the bar for adoption and usability. Most importantly, it is both necessary and beneficial to demonstrate the effectiveness of the proposed method through model replication experiments to validate its feasibility.
> > > >
> > > > Additionally, while federated learning (FL) was mentioned, its applicability to the proposed method remains unclear. In FL, updates from individual clients are typically aggregated on the server side (e.g., via FedAvg), which significantly reduces the influence of any single client’s contributions. Addressing this aspect is essential for understanding the method’s robustness in such scenarios.
> > > >
> > > > Given these unresolved concerns, I believe the paper is not ready for publication at this stage, and I will maintain my current rating.

---

> > > > > ### Author Response · Authors · 2024-11-26
> > > > > **Response to Key Concerns About the Feasibility of Our Approach**
> > > > >
> > > > > Dear Reviewer Ha5f,
> > > > >
> > > > > Thank you for your valuable feedback and suggestions!
> > > > >
> > > > > We also agree that demonstrating the effectiveness of the proposed method through model replication experiments is necessary and beneficial, which is important to prove the feasibility of our method in the more practical black-box scenarios. Therefore, we conducted additional experiments on the black-box setting on MNIST and CIFAR10. In this setting, the unlearning user cannot access the server's current model. The unlearning user only knows the type of the model (MLP on MNIST and CNN on CIFAR10 in our experiment), and the user only has the erasing data and the auxiliary data. We set the size of auxiliary data to 1% of the server-side training data. Other unlearning settings are the same as the main setting in the paper, where we first backdoor the erasing data for model training and aim to unlearn these backdoored erasing data.
> > > > >
> > > > >
> > > > >
> > > > > With these settings, the unlearning user trains a shadow model ($\theta_s$) with 94.55% accuracy on MNIST and 42.57% accuracy on CIFAR10. By contrast, the accuracy of the server's model ($\theta_o$) trained with the entire dataset is 98.74% on MNIST and 78.80% on CIFAR10. Since both models are optimized on the erasing dataset, the proposed efficient unlearning update estimation (EUUE) method is effective for simulating the update of the unlearning data based on the shadow model. Hence, we can generate effective noise for the incremental learning data to approach the influence of unlearning. Then, we upload the constructed data to the server side for incremental learning, aiming to achieve the unlearning effect at the same time.
> > > > >
> > > > >
> > > > > We present the results as follows, which are also presented in **Tab. R6** in the Appendix. J in the revised paper. We also updated the supplementary materials, which separate the newly added context for your convenience. On both datasets, OUbL achieves effective unlearning performance, effectively removing the backdoor influence. The backdoor removal effectiveness in the black-box setting is slightly lower than in the white-box setting. However, the negative impact on the model utility is also mitigated. These experimental results demonstrate the feasibility of OUbL in a more practical scenario, which lets the unlearning user not rely on the assumption of white-box access to the trained model in the federated learning scenarios.
> > > > >
> > > > >
> > > > > The **Table R6** of additional experiments on the black-box setting:
> > > > >
> > > > > | On MNIST       |  USR  = 1%|  2%  | 3%  |  4% | 5% |
> > > > > | --------         | --------    | -------- | -------- |  -------- |   -------- |
> > > > > | Model Acc. (white-box)      | 98.52%      | 98.55%   |  98.15%  |  98.19%   |  95.43%    |
> > > > > | Model Acc. (black-box)     | 98.26%      |  98.20%  |   98.31% |    98.27%  |  98.54%    |
> > > > > | Backdoor Acc. (white-box)    | 9.67%       | 10.08%   | 9.83%   |  10.42%   |  10.57%        |
> > > > > | Backdoor Acc. (black-box)    | 12.33%       |  9.58%   | 11.67%    |  10.64%    |  11.83%        |
> > > > >
> > > > >
> > > > >
> > > > > | On CIFAR10       |  USR  = 1%|  2%  | 3%  |  4% | 5% |
> > > > > | --------         | --------    | -------- | -------- |  -------- |     --------    |
> > > > > | Model Acc. (white-box)     | 73.89%      | 74.57%   |  74.50%  |   75.15%  |     75.99%      |
> > > > > | Model Acc. (black-box)     | 76.06%      | 75.98%  |   74.93%   |    75.06%  |    74.68%      |
> > > > > | Backdoor Acc. (white-box)   | 9.40%       | 7.30%   |  7.87%    |    8.70%  |    7.24%        |
> > > > > | Backdoor Acc. (black-box)   | 13.20%       | 10.20%   |  8.40%    |   10.25%   |    8.28%        |
> > > > >
> > > > >
> > > > > We sincerely hope our diligent efforts can address your concerns.
> > > > >
> > > > >
> > > > > Best regards,
> > > > >
> > > > > Authors

---

> > > > > > ### Author Response · Authors · 2024-12-01
> > > > > > **Reminder on follow-up discussion (one day left before rebuttal extension ends)**
> > > > > >
> > > > > > Dear Reviewer Ha5f,
> > > > > >
> > > > > > We sincerely appreciate the time and effort you've invested in reviewing our manuscript. Your expertise and dedication to the review process are truly invaluable to us.
> > > > > >
> > > > > > As the extended discussion phase draws to a close in one day, we kindly request you to provide your feedback on our rebuttal. Your insights are of immense importance to us, and we look forward to any additional comments you may have. If our replies have addressed your concerns, we would be thankful for your recognition of this. Should there be any points requiring further discussion or explanation, please feel free to contact us. Furthermore, we are fully prepared to engage more to enhance our work during this pivotal stage.
> > > > > >
> > > > > > Best regards,
> > > > > >
> > > > > > Authors

---

### Official Review · Reviewer_EoYb · 2024-11-03

**Soundness:** 3
**Presentation:** 3
**Contribution:** 2
**Rating:** 5
**Confidence:** 3

**Summary:**

This paper formulates the oblivious unlearning by learning (OUbL) problem - users strategically adds noise to an auxiliary dataset such that incrementally training the model based on the auxiliary dataset recreates the effect of unlearning the deleted data. The auxiliary noise is calculated using Hessian based approximation of the unlearning influence and then taking gradient steps which minimize the unlearning objective and the training loss on auxiliary dataset. The performance of OUbL is experimentally compared with federated unlearning approaches and differential privacy approaches.

**Strengths:**

The same learning algorithm can be used for unlearning - this is particularly impactful because the current scope of unlearning algorithms is significantly limited.

**Weaknesses:**

It seems unrealistic that each user in a dataset would have access to clean samples from the data distribution, could you clarify how a user would receive clean samples in practice?

Furthermore, as more unlearning requests come in, the distribution over the dataset is changing, so these clean samples would be coming from a changing distribution, making it even more difficult to obtain clean samples. How would this distribution shift be handled?

Users uploading their data to an ML server which already has access to their data during a deletion request does not seem like a major privacy concern. In what specific scenarios will this create a privacy concern?

The paper lacks theoretical guarantees on the privacy of users after unlearning, for example, in terms of the typical privacy epsilon parameter

**Questions:**

In line 409-411, how are these privacy epsilon values for OUbL being concluded? It’s not clear to me from the figure. I think it would be best to explain how these privacy metrics are being derived, especially since there is no theoretical guarantees on the privacy parameter.

Suggestions

Parenthetical citations should be cited with \citep{}, seems like all of the citations are done as in text citations using \citet{}

line 374: sate -> rate

line 157: is it better to use \approx instead of \equal? it doesn’t seem that exact equality is being achieved

---

> ### Author Response · Authors · 2024-11-20
> **Rebuttal by Authors**
>
> # Rebuttal Part 1
>
> We thank you very much for the very insightful comment. We revised the paper according to your suggestions and provided supplementary materials, which separate the newly added context for your convenience. Below, we provide our point-to-point responses to the comments, denoted by **[W]** for weaknesses and **[Q]** for questions.
>
>
> **Response to W1:** We sincerely appreciate the Reviewer's comment. We supplemented the scenario of our settings. Since our method is based on incremental learning, the continual learning scenario is reasonable, where the users will have new data uploaded to the server for the model's update. We present the detailed scenario discussion as follows.
>
> >**Machine Unlearning Service Scenarios.** To facilitate understanding, we introduce the problem in a Machine Learning as a Service (MLaaS) scenario. In the MLaaS setting, there are two key entities: a ML server that trains models as ML services, and users (data owners) who contribute their data for ML model training. In such scenarios, machine unlearning occurs when users realize that some of their previously contributed samples are private and wish to revoke these data contributions from the trained models.
>
> >**The ML Server's Ability.** We assume the ML server is honest but curious [R1]: while it honestly hosts and provides ML services, including model training and updating, it may still be curious about private information, such as **unlearned data** and **unlearning intentions**, if there are other operations. Informing the server of unlearning intentions to customize unlearning operations is considered a privacy threat because it reveals users' unlearning purposes, potentially enabling the server to prepare targeted unlearning attacks [R1,R2]. Therefore, in our setting, we assume the ML server has only the learning algorithm $\mathcal{A}$ and the model with parameters $\theta$ to meet strict privacy requirements. The ML server will not conduct unlearning operations other than training the model using the learning algorithm $\mathcal{A}$ for model updating.
>
> >Moreover, we assume the ML server does not store the original training data and cannot access the erased data, which should not be exposed to the server again due to privacy concerns. This assumption is reasonable in both real-world and privacy-preserving MLaaS scenarios. In real-world applications, vast amounts of data are generated daily, leading to the need for prompt model updates. Consequently, many models are trained using incremental or continual learning techniques [R3,R4]. Therefore, the server does not retain the entire raw data due to its large size [R5,R6]. In privacy-preserving scenarios, the ML server is restricted from directly accessing private training data from users due to privacy concerns [R7,R8].
>
> >**The Users' Ability.** The training data $D$ was collected from all users and was used to train the model $\theta_o$. The unlearning user has the erased data $D_u \subset D$. To estimate the unlearning update as the target for unlearning noise generation in our method, we assume the unlearning user can access the trained model $\theta_o$, which is a common setting even in many privacy-preserving scenarios such as FL. We assume the unlearning user has auxiliary clean examples $D_a$ so that they can synthesize a new dataset based on it with the unlearning noise, replacing the erased data $D_u$ for achieving the unlearning effect with only incremental learning using the synthesized dataset.
>
> References [R1-R8] are provided at the end of the revised manuscript and the first comment.
>
>
>
> **Response to W2:** We greatly thank the Reviewer's insightful comment. We supplemented that it is reasonable for users to have clean data in continual learning scenarios, as they are the data contributors.
> As more samples are unlearned (deleted), the distribution over the dataset is changing but would not be too significant as the number of unlearned samples is negligible to the size of the whole training dataset. The results in [R15] present that 3.2 million requests for deleting URLs have been issued to Google from 2014 to 2019, which certainly constitutes less than 0.01% of the total URLs Google indexes in the 5 years. We thank you again, and we believe the question is valuable in investigating the unlearning for a large amount of deletion, which is our future work.
>
> [R15]. Bertram, Theo, et al. "Five years of the right to be forgotten." Proceedings of the 2019 ACM SIGSAC Conference on Computer and Communications Security. 2019.

---

> > ### Author Response · Authors · 2024-11-20
> > **Rebuttal Part 2**
> >
> > # Rebuttal Part 2
> >
> > **Response to W3:** We sincerely thank the Reviewer's insightful comment. In the previous supplemented discussion about the scenario, we put our setting in the scenario that the server has not stored and cannot access the original training data, which is reasonable in both the normal and privacy-protected scenarios as too much data is generated in the big data era. Moreover, as the Reviewer pointed out that if the server is malicious and has access to the all original data, we believe protecting the unlearning intention and unlearning samples during unlearning requests is also important. It is because the unlearning intention and requirement will remind the server the unlearned samples contain private information. Otherwise, the server may only store these data and does not realize which parts are private.
> >
> >
> >
> > **Response to W4:** We sincerely appreciate the Reviewer's insightful comment. In this paper, we design the solution to protect the unlearning intentions and data through incremental learning. However, as you commented, we still have not presented the corresponding theoretical privacy protection guarantees. We evaluated the privacy protection effect using reconstruction attacks and compared the effect with the corresponding methods with differential privacy. We demonstrate how much privacy protection effect we achieved according to the method with differential privacy. It is a valuable question, and we will further find the theoretical framework to provide the theoretical guarantees of our method.
> >
> >
> >
> > **Response to Q1:** We greatly thank the Reviewer for this question. In Figure 3, the epsilon value of local differential privacy (LDP) is not for the OUbL method; it is the baseline VBU with the simple LDP mechanism, which we denote at VBU-LDP. We explained the mechanism of how to apply LDP in VBU in lines 361 to 367. We chose the Laplace mechanism to generate the noise and added it directly to the unlearned samples. The experiments aim to test how much privacy protection the OUbL achieved compared to the baseline unlearning method with the general LDP mechanism. We also revised the corresponding introduction from lines 409 to 411 according to your suggestion for better clarity, as follows.
> >
> > >The results demonstrate that OUbL achieves privacy protection like $\epsilon=6$ LDP privacy protection of VBU-LDP on MNIST, similar to $\epsilon=4$ LDP privacy protection of VBU-LDP on CIFAR10 and $\epsilon=4$ LDP privacy protection on CelebA.
> >
> >
> >
> >
> > **Response to Suggestions:** We greatly appreciate the Reviewer's suggestions. We replaced the "\cite" using "\citep" and revised the following to issues as you suggested. We also carefully checked the entire paper to fix these typos.

---

> > > ### Author Response · Authors · 2024-11-25
> > > **Reminder on follow-up discussion (one day left before rebuttal ends)**
> > >
> > > Dear Reviewer EoYb,
> > >
> > > We sincerely appreciate the time and effort you've invested in reviewing our manuscript. Your expertise and dedication to the review process are truly invaluable to us.
> > >
> > > As the discussion phase draws to a close in one day, we kindly request you to provide your feedback on our rebuttal. Your insights are of immense importance to us, and we look forward to any additional comments you may have. If our replies have addressed your concerns, we would be thankful for your recognition of this. Should there be any points requiring further discussion or explanation, please feel free to contact us. Furthermore, we are fully prepared to engage more to enhance our work during this pivotal stage.
> > >
> > > Best regards,
> > >
> > > Authors

---

> > > > ### Author Response · Authors · 2024-12-01
> > > > **Reminder on follow-up discussion (one day left before rebuttal extension ends)**
> > > >
> > > > Dear Reviewer EoYb,
> > > >
> > > > We sincerely appreciate the time and effort you've invested in reviewing our manuscript. Your expertise and dedication to the review process are truly invaluable to us.
> > > >
> > > > As the extended discussion phase draws to a close in one day, we kindly request you to provide your feedback on our rebuttal. Your insights are of immense importance to us, and we look forward to any additional comments you may have. If our replies have addressed your concerns, we would be thankful for your recognition of this. Should there be any points requiring further discussion or explanation, please feel free to contact us. Furthermore, we are fully prepared to engage more to enhance our work during this pivotal stage.
> > > >
> > > > Best regards,
> > > >
> > > > Authors

---

### Official Review · Reviewer_5Whi · 2024-11-04

**Soundness:** 3
**Presentation:** 2
**Contribution:** 3
**Rating:** 6
**Confidence:** 3

**Summary:**

The paper addresses a limitation in current unlearning methods, which require notifying the server to apply the unlearning algorithm—potentially leading to privacy leakage. To address this, it introduces a new method called OUbL. OUbL constructs a noisy dataset that enables the model to unlearn specific data using its original algorithm, without explicit server intervention. Experimental results indicate that this method performs well in preserving privacy.

**Strengths:**

1. The paper introduces an interesting perspective, highlighting that current unlearning methods often require informing the server about the data to be forgotten, which could pose privacy risks.
2. The proposed approach is clearly explained and well-motivated.

**Weaknesses:**

1. Existing work addresses privacy issues for the forgetting set using membership inference (MI) attacks as a common metric. Comparing your approach with these methods using MI attacks could strengthen the paper.

References: Kurmanji, M., Triantafillou, P., Hayes, J. and Triantafillou, E., 2024. Towards unbounded machine unlearning. Advances in neural information processing systems, 36.

2. The method requires continuous access to the training set. It is unclear how it would function if access to the training set were unavailable.


Presentations:
1. In-text citations lack parentheses. For example, on line 42, it should be written as (Thudi et al., 2022; Hu et al., 2024b) instead of Thudi et al. (2022); Hu et al. (2024b).
2. Some abbreviations are not clearly explained. For instance, it is difficult to understand what "USR" refers to on line 373 until it is defined on line 511.

**Questions:**

The model performance drops by approximately 5% compared to other benchmarks when USR is set to 1% in the experiment. This suggests that the method may impair model performance, especially as the USR increases. Any thoughts on addressing this issue?

---

> ### Author Response · Authors · 2024-11-20
> **Rebuttal by Authors**
>
> # Rebuttal Part 1
>
> We greatly appreciate your insightful comments that precisely recognize the strengths of our work. We additionally conducted experiments with the MI metric, as you suggested, based on the referenced paper. We provided supplementary materials, which separate the newly added context for your convenience. Below, we offer our detailed responses to your comments, categorized by **[W]** for weaknesses and **[Q]** for questions.
>
>
> **Response to W1:** We sincerely thank the Reviewer's suggestion. We conducted additional experiments using the membership inference (MI) attack metric based on the referenced paper [R12] and [R13]. As [R12], we train a binary classifier on the unlearned model's loss on erased and test examples for the objective of classifying "in" (forget) versus "out" (test). Then, we make predictions for held-out losses (losses that weren't trained on) that are balanced between forget and test losses. For a perfect defense, the MI accuracy is 50%, indicating that it is unable to separate the two sets, marking a success for the unlearning method. We demonstrate the corresponding results in the following table. We set the unlearning sample rate (USR) equal to 1% and test various parameters of auxiliary sample rate (ASR) and clean sample rate (CSR). The results demonstrate that larger ASR and CSR ensure a better unlearning effect, i.e., lower MI accuracy.
>
> We also want to explain why we choose the membership inference via backdoor (MIB metric, the backdoor accuracy). MIB [R14] infers the unlearning effect based on the removal effect of backdoored samples. The disappearance of the influence of backdoor samples can directly demonstrate the unlearning effect, which we believe is more straightforward than MI. The backdoor accuracy of MIB usually drops from 100\% to 10\%, while the accuracy of MI usually drops from 60\% to 50\%.
>
> [R12]. Kurmanji, M., Triantafillou, P., Hayes, J. and Triantafillou, E., 2024. Towards unbounded machine unlearning.Advances in neural information processing systems, 36.
>
> [R13]. Chen, Min, et al. "When machine unlearning jeopardizes privacy." Proceedings of the 2021 ACM SIGSAC conference on computer and communications security. 2021.
>
> [R14]. Hu, Hongsheng, et al. "Membership inference via backdooring." IJCAI (2022).
>
>
> The **Table R3** of Membership inference attack accuracy:
>
> | Dataset          | Original    |  ASR and CSR, 1%  | 2%        |  3%       |    4%       |
> | --------         | --------    | --------          | --------  | --------  | --------    |
> |  On MNIST        | 63.86%      | 53.87%            |  53.61%   |   53.02%  |    52.86%   |
> |  On CIFAR10      | 77.43%      | 61.47%            |  61.30%   |   61.10%  |    60.92%   |
> |  On CelebA       | 58.37%      | 51.94%            | 51.32%    |  51.04%   |    50.67%   |
>
>
>
> **Response to W2:** We appreciate the Reviewer's comment. There may be some unclarity in our presentation of the method. In fact, our approach does not require access to the training set. Eq.(4) is the ideal situation that we hope to achieve but is intractable. We relaxed the problem and solved it from lines 256 to 268. As suggested by Reviewer BZGC, we replaced the original model $\theta$ using $\theta_o$ for clarity. All our methods required are the original model $\theta_o$, the unlearned data $D_u$, and some auxiliary data $D_a$. We present the revised method introduction (lines 256 to 268), detailed scenario setting and requirements as follows.
>
> >**Method introduction (Lines 256 to 268):** Unfortunately, calculating $\Delta^p$ that satisfies Eq.(5) is also intractable as it is required to hold for all values of $\theta$. In our setting, the unlearning user cannot access $\theta$ for samples in the remaining dataset $D \setminus D_u$. One possible solution proposed in (Geiping et al. 2021; Di et al. 2022) is to relax Eq.(5) to be satisfied for a fixed model — the model obtained by training on the original dataset. We assume a well-trained model $$\theta_o$$ before unlearning and fix it during unlearning noise generation. Then, we can minimize the loss based on the cosine similarity between the two gradients as:
>
> >$$\phi (\Delta^p, \theta_o) = 1 - \frac{\langle \Delta \theta_{D_u}, -\frac{1}{P} \sum_{i=1}^P \nabla_{\theta} \ell ((x_i + \Delta_i^p,y_i);\theta_o) \rangle}{ \|  \Delta \theta_{D_u} \| \cdot  \|  -\frac{1}{P}\sum_{i=1}^P \nabla_{\theta} \ell ((x_i + \Delta_i^p,y_i);\theta_o) \|}.$$
>
> >(Geiping et al. 2021) use $R$ restarts, usually $R \leq 10$, to increase the robustness of noise synthesis. Using this scheme, we can also find suitable noise for unlearning. However, it is not always effective because we cannot always achieve satisfactory random noise within $10$ restarts. To address this issue, we propose an unlearning noise descent strategy.
>
>
> Scenarios and threat model are attached in the next part.

---

> ### Author Response · Authors · 2024-11-20
> **Rebuttal Part 2**
>
> # Rebuttal Part 2
>
> >**Machine Unlearning Service Scenarios.** To facilitate understanding, we introduce the problem in a Machine Learning as a Service (MLaaS) scenario. In the MLaaS setting, there are two key entities: a ML server that trains models as ML services, and users (data owners) who contribute their data for ML model training. In such scenarios, machine unlearning occurs when users realize that some of their previously contributed samples are private and wish to revoke these data contributions from the trained models.
>
> >**The ML Server's Ability.** We assume the ML server is honest but curious [R1]: while it honestly hosts and provides ML services, including model training and updating, it may still be curious about private information, such as **unlearned data** and **unlearning intentions**, if there are other operations. Informing the server of unlearning intentions to customize unlearning operations is considered a privacy threat because it reveals users' unlearning purposes, potentially enabling the server to prepare targeted unlearning attacks [R1,R2]. Therefore, in our setting, we assume the ML server has only the learning algorithm $\mathcal{A}$ and the model with parameters $\theta$ to meet strict privacy requirements. The ML server will not conduct unlearning operations other than training the model using the learning algorithm $\mathcal{A}$ for model updating.
>
> >Moreover, we assume the ML server does not store the original training data and cannot access the erased data, which should not be exposed to the server again due to privacy concerns. This assumption is reasonable in both real-world and privacy-preserving MLaaS scenarios. In real-world applications, vast amounts of data are generated daily, leading to the need for prompt model updates. Consequently, many models are trained using incremental or continual learning techniques [R3,R4]. Therefore, the server does not retain the entire raw data due to its large size [R5,R6]. In privacy-preserving scenarios, the ML server is restricted from directly accessing private training data from users due to privacy concerns [R7,R8].
>
> >**The Users' Ability.** The training data $D$ was collected from all users and was used to train the model $\theta_o$. The unlearning user has the erased data $D_u \subset D$. To estimate the unlearning update as the target for unlearning noise generation in our method, we assume the unlearning user can access the trained model $\theta_o$, which is a common setting even in many privacy-preserving scenarios such as FL. We assume the unlearning user has auxiliary clean examples $D_a$ so that they can synthesize a new dataset based on it with the unlearning noise, replacing the erased data $D_u$ for achieving the unlearning effect with only incremental learning using the synthesized dataset.
>
>
> References [R1-R8] are provided at the end of the revised manuscript and the first comment.
>
>
>
> **Response to Presentations 1:** We thank the Reviewer's suggestion, which is also suggested by Reivewer BZGC. We replaced most "\cite" using "\citep".
>
>
> **Response to Presentations 2:** We thank the Reviewer's comment. We revised this abbreviations, and we carefully checked the whole paper as you suggested.
>
> **Response to Questions:** We sincerely appreciate the Reviewer's question. Our method does impair the model utility to some extent. Therefore, we propose the compensation method by uploading some clean samples together. Moreover, with a larger auxiliary sample rate and clean sample rate, we can effectively mitigate the model utility degradation, which is presented in Figure 6 in the original manuscript and Figure 5 in the revised manuscript. Besides increasing the auxiliary and clean sample rates, we consider that dividing the unlearning requests into different uploading rounds may also mitigate the model utility degradation. It is valuable for the future investigation. We thank you for your question again.

---

> > ### Author Response · Authors · 2024-11-25
> > **Reminder on follow-up discussion (one day left before rebuttal ends)**
> >
> > Dear Reviewer 5Whi,
> >
> > We sincerely appreciate the time and effort you've invested in reviewing our manuscript. Your expertise and dedication to the review process are truly invaluable to us.
> >
> > As the discussion phase draws to a close in one day, we kindly request you to provide your feedback on our rebuttal. Your insights are of immense importance to us, and we look forward to any additional comments you may have. If our replies have addressed your concerns, we would be thankful for your recognition of this. Should there be any points requiring further discussion or explanation, please feel free to contact us. Furthermore, we are fully prepared to engage more to enhance our work during this pivotal stage.
> >
> > Best regards,
> >
> > Authors

---

> > > ### Author Response · Authors · 2024-12-01
> > > **Reminder on follow-up discussion (one day left before rebuttal extension ends)**
> > >
> > > Dear Reviewer 5Whi,
> > >
> > > We sincerely appreciate the time and effort you've invested in reviewing our manuscript. Your expertise and dedication to the review process are truly invaluable to us.
> > >
> > > As the extended discussion phase draws to a close in one day, we kindly request you to provide your feedback on our rebuttal. Your insights are of immense importance to us, and we look forward to any additional comments you may have. If our replies have addressed your concerns, we would be thankful for your recognition of this. Should there be any points requiring further discussion or explanation, please feel free to contact us. Furthermore, we are fully prepared to engage more to enhance our work during this pivotal stage.
> > >
> > > Best regards,
> > >
> > > Authors

---

### Official Review · Reviewer_iCGz · 2024-11-04

**Soundness:** 3
**Presentation:** 3
**Contribution:** 3
**Rating:** 8
**Confidence:** 4

**Summary:**

This paper proposes a method for machine unlearning through learning. The idea is to synthesize an auxiliary dataset, whose gradients match the model updates caused by the data to be unlearned. In this way, the updates due to the auxiliary dataset will cancel out the updates due to the data being deleted. In the meantime, the server does not know which records the user wanted to delete in the first place, providing an extra layer of privacy protection (avoiding secondary injury when curing the initial injury).

**Strengths:**

1. The idea of oblivious unlearning–do not let the server know which records the user wants to delete in the first place is a crucial concern in unlearning, and has not been addressed in previous work.
2. The proposed solution is a novel application of gradient matching in the field of machine unlearning. Overall the idea makes sense and the solution is very elegant. I do have some questions regarding the practicality of the proposed solution and I will consider raising my rating if all are resolved. Please refer to the weakness section.

**Weaknesses:**

1. The setup that the server chooses to continue to learn on the provided auxiliary data is not motivated. E.g., the server may stop training before the user realizes that she/he wants to delete some data from the model, and the unlearning algorithm is never run. Hence, it is not entirely correct that the intention of unlearning can be hidden. In addition, if the server himself is malicious, then he sees the data to be deleted from the very beginning of the training process. Machine unlearning also does not help, since privacy is breached before unlearning happens. If the server is benign, and other end users of the model are the malicious ones, then there is no need for oblivious unlearning (since the server is benign).

2. How to estimate the model change caused by the data being deleted seems to be a difficult problem, without the knowledge of the training algorithm and the training dataset. For the training dataset, the authors assume that they have access to some training data. Is there any constraint on the size and distribution of this small dataset, e.g., how large it is compared to the dataset to be deleted, and the whole training dataset, so that the proposed solution could work? In addition, if the training algorithm is unknown (maybe also the learning rate and optimizer are not known), then how to ensure the overall changes to be applied (caused by the constructed auxiliary dataset) is approximately the same as the change caused by the data to be deleted. E.g., if the learning rate is large and causes the auxiliary data’s gradients to overshoot the original changes to be canceled out, then the privacy of the data to be deleted cannot be preserved.

3. The idea of gradient matching is similar to a previous paper in the field of private ML, Deep Leakage from Gradients by Zhu et al. I am curious what would happen if the user directly applies their method to construct the auxiliary dataset. Would the method be no longer oblivious (the constructed dataset may look weird)?

4. From the server’s perspective, how to distinguish a benign user, who wants to delete their data from a malicious user, who wants to upload noisy gradients to sabotage the model performance?

5. Can you provide some examples (high-resolution preferred) of the original auxiliary dataset and their noisy counterparts?

**Questions:**

Refer to the weakness section.

---

> ### Author Response · Authors · 2024-11-20
> **Rebuttal by Authors**
>
> # Rebuttal Part 1
>
>
> We greatly thank Reviewer iCGz for acknowledging the contributions, soundness, and presentation quality of our paper. And we sincerely appreciate Reviewer iCGz for proposing these insightful questions. We uploaded the revised manuscript and provided supplementary materials, which separate the newly added context for your convenience. Below, we provide our responses to the comments, denoted by [W] for weaknesses.
>
>
> **Response to W1:** We greatly appreciate the Reviewer's insightful comment. We supplemented discussion about the scenario of our setting. We set the scenario in continual learning, which would be motivated that the server will continually update the model, and the users will upload the new data for model update [R3, R4]. We assume the ML server is honest but curious [R1]: while it honestly hosts and provides ML services, including model training and updating, it may still be curious about private information, such as **unlearned data** and **unlearning intentions**, if there are other operations. Moreover, we assume the ML server does not store the original training data and cannot access the erased data, which should not be exposed to the server again due to privacy concerns. Additionally, as the Reviewer pointed out that if the server is malicious and has access to the all original data, we believe protecting the unlearning intention and unlearning samples during unlearning requests is also important. It is because the unlearning intention and requirement will remind the server the unlearned samples contain private information. Otherwise, the server may only store these data and does not realize which parts are private. We demonstrate the scenario discussion as follows.
>
>
> >**Machine Unlearning Service Scenarios.** To facilitate understanding, we introduce the problem in a Machine Learning as a Service (MLaaS) scenario. In the MLaaS setting, there are two key entities: a ML server that trains models as ML services, and users (data owners) who contribute their data for ML model training. In such scenarios, machine unlearning occurs when users realize that some of their previously contributed samples are private and wish to revoke these data contributions from the trained models.
>
> >**The ML Server's Ability.** We assume the ML server is honest but curious [R1]: while it honestly hosts and provides ML services, including model training and updating, it may still be curious about private information, such as **unlearned data** and **unlearning intentions**, if there are other operations. Informing the server of unlearning intentions to customize unlearning operations is considered a privacy threat because it reveals users' unlearning purposes, potentially enabling the server to prepare targeted unlearning attacks [R1,R2]. Therefore, in our setting, we assume the ML server has only the learning algorithm $\mathcal{A}$ and the model with parameters $\theta$ to meet strict privacy requirements. The ML server will not conduct unlearning operations other than training the model using the learning algorithm $\mathcal{A}$ for model updating.
>
> >Moreover, we assume the ML server does not store the original training data and cannot access the erased data, which should not be exposed to the server again due to privacy concerns. This assumption is reasonable in both real-world and privacy-preserving MLaaS scenarios. In real-world applications, vast amounts of data are generated daily, leading to the need for prompt model updates. Consequently, many models are trained using incremental or continual learning techniques [R3,R4]. Therefore, the server does not retain the entire raw data due to its large size [R5,R6]. In privacy-preserving scenarios, the ML server is restricted from directly accessing private training data from users due to privacy concerns [R7,R8].
>
> >**The Users' Ability.** The training data $D$ was collected from all users and was used to train the model $\theta_o$. The unlearning user has the erased data $D_u \subset D$. To estimate the unlearning update as the target for unlearning noise generation in our method, we assume the unlearning user can access the trained model $\theta_o$, which is a common setting even in many privacy-preserving scenarios such as FL. We assume the unlearning user has auxiliary clean examples $D_a$ so that they can synthesize a new dataset based on it with the unlearning noise, replacing the erased data $D_u$ for achieving the unlearning effect with only incremental learning using the synthesized dataset.

---

> > ### Author Response · Authors · 2024-11-20
> > **Rebuttal Part 2**
> >
> > #Rebuttal Part 2
> >
> > **Response to W2:** We sincerely thank the Reviewer's insightful comment. We agree that estimate the influence of unlearned data without the knowledge of the training algorithm and the training dataset is difficult. Hence, we propose Efficient unlearning update estimation (EUUE) method in Section 3.2, as shown in Algorithm 1 in Appendix B. Our method would not rely on the training dataset, but we need to know the learning algorithm. The unlearning user also needs to have unlearned data and some clean samples that have the same distribution as the training dataset. We believe it is achievable in many settings, even in privacy-protecting scenarios such as federated learning. The users in federated learning will know the learning algorithm and have the data to assist in global model training. For the unlearning user himself/herself, if he/she has a larger auxiliary data set, he/she can use more data to construct the dataset for unlearning, ensuring a better unlearning effect, as shown in Figure 7.
> >
> > Additionally, we conduct experiment to illustrate the influence of different learning rate using our synthesized data.
> > The results are presented as follows, which demonstrate that a larger learning rate can speed the convergence to achieve unlearning, costing less computation and achieving a better unlearning effect (low backdoor accuracy by removing). The tradeoff is that it slightly decreases the model utility at the same time, which is not too much on MNIST but a little worse on CIFAR10.
> >
> >
> >
> >
> > The **Table R2** of evaluating learning rate on MNIST and CIFAR10: USR=1%, ASR=1%, and CSR=1%:
> >
> > | On MNIST       | Learning Rate: 0.0001|  0.0002  | 0.0004  |  0.0006 | 0.0008 |
> > | --------         | --------    | -------- | -------- |  -------- |   -------- |
> > | Model Acc.       | 98.52%      | 97.84%   |  96.72%  |  95.88%   |  95.37%    |
> > | Backdoor Acc.    | 9.67%       | 9.53%   | 9.17%   |  8.20%   |  8.67%     |
> > | Running Time     | 3.92        | 2.72     | 1.83     |  1.61     |  1.56      |
> >
> >
> > | On CIFAR10       | Learning Rate: 0.0001|  0.0002  | 0.0004  |  0.0006 |    0.0008 |
> > | --------         | --------    | -------- | -------- |  -------- |     --------    |
> > | Model Acc.       | 73.89%      | 72.98%   |  68.69%  |   65.23%  |     62.23%      |
> > | Backdoor Acc.    | 9.40%       | 6.20%   |  5.80%    |    4.00%  |    2.48%        |
> > | Running Time     | 6.63        | 3.72     | 2.83     |  2.51     |    2.23         |
> >
> >
> >
> > **Response to W3:** We sincerely appreciate the Reviewer's insightful comment for pointing out similar solutions. If we directly synthesize the dataset for unlearning by matching the gradient to the estimated unlearning influence, it can also implement unlearning, but as you said, the constructed dataset may look weird, not like the genuine samples. We present some examples in Figure R1 in Supplementary Material, which directly applied [R13] and is demonstrated in the same figure as the **Response to W5**.
> >
> >
> > **Response to W4:** We greatly thank the Reviewer's insightful question. This is a great question and is worth investigating further in future work. Now, we can only propose some possible ways for the server to distinguish these two kinds of users. The most significant difference is the purposes of the unlearning user and the malicious user. Unlearning users want to remove some knowledge of their data from the model, and they also want to preserve the model's utility. Therefore, most clean samples and the auxiliary data they choose are in the same distribution as the genuine samples, and the synthesized noise should not influence the utility of the remaining dataset, as shown in the second objective of Eq.(4). However, the purpose of the malicious user is to sabotage the model performance. Their uploaded data will not be consistent with the genuine samples, so they can degrade model utility. We believe checking the similarity between the uploaded samples and genuine samples would be a possible solution to this question. However, detailed poisoning attacking methods may need different solutions, and the question is valuable to investigate in future work. We also supplemented the discussion in the manuscript in Appendix.H.
> >
> >
> >
> >
> > **Response to W5:** We sincerely thank the Reviewer's insightful comments. We illustrate some examples in **Figure R1** in Supplementary Material, which demonstrates the original auxiliary dataset (the first row), their noisy counterparts (the middle row), and directly construct data without an auxiliary dataset [R11] (the last row).
> >
> > [R11]. Zhu, Ligeng, Zhijian Liu, and Song Han. "Deep leakage from gradients." Advances in neural information processing systems 32 (2019).

---

> > > ### Author Response · Authors · 2024-11-25
> > > **Reminder on follow-up discussion (one day left before rebuttal ends)**
> > >
> > > Dear Reviewer iCGz,
> > >
> > > We sincerely appreciate the time and effort you've invested in reviewing our manuscript. Your expertise and dedication to the review process are truly invaluable to us.
> > >
> > > As the discussion phase draws to a close in one day, we kindly request you to provide your feedback on our rebuttal. Your insights are of immense importance to us, and we look forward to any additional comments you may have. If our replies have addressed your concerns, we would be thankful for your recognition of this. Should there be any points requiring further discussion or explanation, please feel free to contact us. Furthermore, we are fully prepared to engage more to enhance our work during this pivotal stage.
> > >
> > > Best regards,
> > >
> > > Authors

---

> > > > ### Comment · Reviewer_iCGz · 2024-11-25
> > > > **My concerns are addressed**
> > > >
> > > > Thanks for the response. My concerns are fully addressed. Therefore I have raised my score.
> > > >
> > > > Please incorporate the changes into the final version of your paper (consider moving some of them to the main part). They are very important for future follow-up works, including clearly stating the problem definition (responses to W1 and W4) and the motivation for the auxiliary dataset (W3). I hope the community will benefit from your interesting findings

---

> > > > > ### Author Response · Authors · 2024-11-25
> > > > > **Thank You!**
> > > > >
> > > > > Dear Reviewer iCGz,
> > > > >
> > > > > Thank you for your response and your valuable input in enhancing our submission. We are pleased to hear that our responses have been satisfactory. We will further revise our paper according to your suggestions.
> > > > >
> > > > > Best regards,
> > > > >
> > > > > Authors

---

### Official Review · Reviewer_BZGC · 2024-11-05

**Soundness:** 2
**Presentation:** 2
**Contribution:** 2
**Rating:** 6
**Confidence:** 3

**Summary:**

The paper studies the problem of implementing machine unlearning without exposing erased data and unlearning intentions to the server. The authors propose an Oblivious Unlearning by Learning (OUbL) approach to protect both unlearned data and unlearning intentions during machine unlearning processes. OUbL constructs a new dataset with synthesized unlearning noise and achieves unlearning by incremental learning. Comprehensive experiments demonstrate the effectiveness of OUbL.

**Strengths:**

1. The paper is the first to identify the privacy threats posed by the exposure of both unlearned data and unlearning intentions during machine unlearning processes. The idea is novel.
2. The experiments are extensive and show the significant superiority of OUbL over SOTAs in terms of both privacy protection and unlearning effectiveness.
3. The paper is overall well-structured. The narrative is easy to follow.

**Weaknesses:**

1. Issues with the writing.
   - Lack of clarification on notations. E.g., the mapping of the functions $\mathcal U(\cdot)$ and $\mathcal A(\cdot)$ in the Problem Statement part, the definition of  $\nabla$ and the function $\ell$ in Section 3.1, and the definition of $\mathcal I$ in Eq.(3), the description of $I$ on line 225.
   - Some typos, e.g., missing $\nabla$ on line 221.
   - The $H_\theta^{-1}$ in Eq.(3) denotes the inverse of the Hessian matrix evaluated at $\theta$ **on the dataset $D\backslash D_u$**.
   - The reviewer suggests using another symbol like $\theta_o$ instead of simply $\theta$ to represent the original trained model for reducing ambiguity since $\theta$ seems to be a variable in Eq.(4).
   - The reviewer suggests using the command "\citep" instead of "\cite" when the references are not the objectives in the sentence.
   - Presenting Algorithm 2 with a detailed description in the main text would be better.
2. Issues with the figures.
   - Both Figure 1 and Figure 2 describe the main process of OUbL, with Figure 2 being more detailed. Therefore, the reviewer thinks that Figure 1 is unnecessary.
   - Incomplete compilation of notation $\mathcal I_{D_u}$ in Figure 2.
   - The Problem Statement of OUbL on page 3 doesn't include the phase of Figure 2 c), the incremental learning step with clean dataset $D_c$.
3. Inconsistencies between Eq.(4) and Eq.(8). The second objective in Eq.(4) is evaluated on the dataset $D\backslash D_u$ (size $N$) and Eq.(8) is evaluated on the dataset $D_a$ (size $P$).
4. The reviewer thinks that the comparisons between centralized unlearning methods and federated unlearning methods are unfair since the settings are different. The author should discuss the performances of OUbL applied in federated unlearning scenarios.

**Questions:**

1. What does CSR represent? The authors use SCR to represent the Constructed Samples Rate in Section 4.2 but use it to represent the Clean Samples Rate in Section 4.4.
2. The reviewer is confused about how the second objective in Eq.(4) is satisfied.
3. How do you obtain the datasets $D_a$ and $D_c$? What if the samples in $D_c$ have similar features to the unlearned data? Please add discussions in the main text.

---

> ### Author Response · Authors · 2024-11-20
> **Rebuttal by Authors**
>
> # Rebuttal Part 1
>
> We greatly appreciate Reviewer BZGC for providing a detailed summary of our strengths. And greatly appreciate Reviewer BZGC for proposing these insightful questions and providing these constructive suggestions. We have uploaded the manuscript, which has been revised according to your suggestions. We also provided supplementary materials, which separate the newly added context for your convenience. Below, we provide our responses to the comments, denoted by [W] for weaknesses and [Q] for questions.
>
>
> **Response to W1:** We greatly appreciate Reviewer BZGC for your patience in pointing out these writing issues. We have fixed all the pointed issues. We summarize the changes as below.
>
> - After the problem statement, we add explanation about $\theta_{unl} \gets \mathcal{U}(\cdot) $ and $ \theta_{inc} \gets \mathcal{A}(\cdot) $, which denotes the functions with inputs of model parameters and data, and output the unlearned model and incremental learned model. In Section 3.1, we add the explanation about $ \nabla $ and $\ell$, which denotes the gradients and the loss function of the learning algorithm. In Eq.(3), $\mathcal{I}^{(1)}(D_u)$ denotes the estimated first-order influence of the unlearning data $D_u$. We also supplemented in line 225 $I$ is the identity matrix.
> - We fixed the typo in line 221, and carefully checked the entire paper.
> - We revised the sentence as suggested.
> - We thank the Reviewer's suggestion, and we revised the corresponding parts of the entire paper using $ \theta_o $ instead of $\theta$ to represent the original trained model, including the parts in Appendix.
> - We thank the Reviewer's suggestion, and we replaced most "\cite" using "\citep".
> - We agree that Algorithm 2 is an important contribution to this paper. Therefore, as you suggested, we put it in the main text and added a detailed description as follows.
> >Algorithm 2 synthesizes unlearning noise to create a perturbed dataset $D_{a,p} = (X_a + \Delta^p, Y_a)$. Firstly, we generate a noise matrix $\Delta^{p}$ as shown in line 1 of Algorithm 2 and treat it as parameters that could be updated during optimization. Then, during optimization, we fix the trained model parameters $\theta$ and add the noise to the auxiliary data $D_a$ as the input to the model (line 4). We calculate the gradients of the noise-synthesized data based on the current model point but do not update the model (line 5). Moreover, we calculate the minimization loss according to Eq.(6) (line 6). With this loss, we can use the gradient descent method for both the model and the unlearning noise matrix, but we only update the noise matrix $\Delta^{p}$ while keeping the model $\theta$ fixed (line 7). After a few rounds of iteration, we can synthesize sufficient unlearning noise to the auxiliary data to achieve the unlearning effect.
>
> **Response to W2:** We sincerely thank Reviewer BZGC to provide the constructive suggestions about figures. We response these comments as follows.
>
> - We appreciated the Reviewer's suggestion. We revised the paper as you suggested by removing the original Figure 1 in the Introduction section. We also revised the corresponding introduction parts.
> - We revised the $\mathcal{I}_{D_u} $ in Figure 2.
> - The problem statement is the idea station to achieve oblivious unlearning by learning. Ideally, we can directly achieve both the unlearning effect and model utility preservation to solve the problem. In practice, it will cause the model utility degradation. Therefore, we additionally employ clean data to mitigate the model utility degradation. It is a compensation for our solution; hence, we have not added it to the problem statement.

---

> ### Author Response · Authors · 2024-11-20
> **Rebuttal Part 2**
>
> # Rebuttal Part 2
>
> **Response to W3:** We are appreciated of the Reviewer's this comment. After your former suggestion of replacing $\theta$ using $\theta_o$ to denote the original trained model, which is revised in the manuscript.
>
> Eq.(4) is the ideal and theoretical station to solve the oblivious unlearning problem, i.e., the unlearning users can access the full training data $D$, so they can achieve the retrained model. However, in our setting, the unlearning users cannot access $\theta$ for samples in the remaining dataset $ D \setminus D_u $. Our solution is to relax the second objective of Eq.(4) to be satisfied with a fixed model. And we choose the model obtained by the training on the original dataset, $\theta_o$. It is reasonable for the following reasons: (1) our aim is achieving unlearning by incremental learning, and it is reasonable that the server will continually learn based on the trained model $\theta_o$ rather than an initial model; (2) continually learning for $\theta_o$ based on $D_a^p$ with the first objective in Eq.(4) will simultaneously meet the unlearning purpose of $D_u$ and incremental learning purpose of $D_a^p$. The corresponding introduction of these explanation is presented from line 256 to line 267 in the manuscript. The Eq.(8) is the relaxed incremental learning objective which is based on $\theta_o$. We also attached the revised line 256 to 269 as follows.
>
> >Unfortunately, calculating $\Delta^p$ that satisfies Eq.(5) is also intractable as it is required to hold for all values of $\theta$. In our setting, the unlearning user cannot access $\theta$ for samples in the remaining dataset $D \setminus D_u$. One possible solution proposed in (Geiping et al. 2021; Di et al. 2022) is to relax Eq.(5) to be satisfied for a fixed model — the model obtained by training on the original dataset. We assume a well-trained model $\theta_o$ before unlearning and fix it during unlearning noise generation. Then, we can minimize the loss based on the cosine similarity between the two gradients as:
>
> >$$\phi (\Delta^p, \theta_o) = 1 - \frac{\langle \Delta \theta_{D_u}, -\frac{1}{P} \sum_{i=1}^P \nabla_{\theta} \ell ((x_i + \Delta_i^p,y_i);\theta_o) \rangle}{ \|  \Delta \theta_{D_u} \| \cdot  \|  -\frac{1}{P}\sum_{i=1}^P \nabla_{\theta} \ell ((x_i + \Delta_i^p,y_i);\theta_o) \|}.$$
>
> >(Geiping et al. 2021) use $R$ restarts, usually $R \leq 10$, to increase the robustness of noise synthesis. Using this scheme, we can also find suitable noise for unlearning. However, it is not always effective because we cannot always achieve satisfactory random noise within $10$ restarts. To address this issue, we propose an unlearning noise descent strategy.

---

> > ### Author Response · Authors · 2024-11-20
> > **Rebuttal Part 3**
> >
> > # Rebuttal Part 3
> > **Response to W4:** We sincerely thank the Reviewer's comment. We added the discussion about the threat model as follows, including the server's ability and the users' ability. We also supplemented the discussion in our revised manuscript in Appendix.G. Since our unlearning preparation is on the user side, and the execution of incremental learning is on the server side, the server only conducts the continual learning without the need to access the original data and erased data. We believe it is also feasible in current federated unlearning scenarios, especially frameworks that put unlearning execution on the server side [R9, R10].
> >
> > >**Machine Unlearning Service Scenarios.** To facilitate understanding, we introduce the problem in a Machine Learning as a Service (MLaaS) scenario. In the MLaaS setting, there are two key entities: a ML server that trains models as ML services, and users (data owners) who contribute their data for ML model training. In such scenarios, machine unlearning occurs when users realize that some of their previously contributed samples are private and wish to revoke these data contributions from the trained models.
> >
> > >**The ML Server's Ability.** We assume the ML server is honest but curious [R1]: while it honestly hosts and provides ML services, including model training and updating, it may still be curious about private information, such as **unlearned data** and **unlearning intentions**, if there are other operations. Informing the server of unlearning intentions to customize unlearning operations is considered a privacy threat because it reveals users' unlearning purposes, potentially enabling the server to prepare targeted unlearning attacks [R1,R2]. Therefore, in our setting, we assume the ML server has only the learning algorithm $\mathcal{A}$ and the model with parameters $\theta$ to meet strict privacy requirements. The ML server will not conduct unlearning operations other than training the model using the learning algorithm $\mathcal{A}$ for model updating.
> >
> > >Moreover, we assume the ML server does not store the original training data and cannot access the erased data, which should not be exposed to the server again due to privacy concerns. This assumption is reasonable in both real-world and privacy-preserving MLaaS scenarios. In real-world applications, vast amounts of data are generated daily, leading to the need for prompt model updates. Consequently, many models are trained using incremental or continual learning techniques [R3,R4]. Therefore, the server does not retain the entire raw data due to its large size [R5,R6]. In privacy-preserving scenarios, the ML server is restricted from directly accessing private training data from users due to privacy concerns [R7,R8].
> >
> >
> > >**The Users' Ability.** The training data $D$ was collected from all users and was used to train the model $\theta_o$. The unlearning user has the erased data $D_u \subset D$. To estimate the unlearning update as the target for unlearning noise generation in our method, we assume the unlearning user can access the trained model $\theta_o$, which is a common setting even in many privacy-preserving scenarios such as FL. We assume the unlearning user has auxiliary clean examples $D_a$ so that they can synthesize a new dataset based on it with the unlearning noise, replacing the erased data $D_u$ for achieving the unlearning effect with only incremental learning using the synthesized dataset.
> >
> > [R1]. Hu, Hongsheng, et al. "Learn what you want to unlearn: Unlearning inversion attacks against machine unlearning." IEEE Symposium on Security and Privacy (SP) (2024).
> >
> > [R2]. Chen, Min, et al. "When machine unlearning jeopardizes privacy." Proceedings of the 2021 ACM SIGSAC conference on computer and communications security. 2021.
> >
> > [R3]. Rolnick, David, et al. "Experience replay for continual learning." Advances in neural information processing systems 32 (2019).
> >
> > [R4]. Lopez-Paz, David, and Marc'Aurelio Ranzato. "Gradient episodic memory for continual learning." Advances in neural information processing systems 30 (2017).
> >
> > [R5]. Wu, Yue, et al. "Large scale incremental learning." Proceedings of the IEEE/CVF conference on computer vision and pattern recognition. 2019.
> >
> > [R6]. Wang, Zifeng, et al. "Learning to prompt for continual learning." Proceedings of the IEEE/CVF conference on computer vision and pattern recognition. 2022.
> >
> > [R7]. Naseri, Mohammad, Jamie Hayes, and Emiliano De Cristofaro. "Local and central differential privacy for robustness and privacy in federated learning." NDSS (2022).
> >
> > [R8]. Bonawitz, Keith, et al. "Practical secure aggregation for privacy-preserving machine learning." proceedings of the 2017 ACM SIGSAC Conference on Computer and Communications Security. 2017.

---

> > > ### Author Response · Authors · 2024-11-20
> > > **Rebuttal Part 4**
> > >
> > > # Rebuttal Part 4
> > > **Response to Q1:** We sincerely thank the Reviewer to point out the issue. CSR in this paper is the clean samples rate. Since the constructed sample is based on the auxiliary sample, we directly using the auxiliary samples rate (ASR), which is the same as the constructed samples rate.
> > >
> > >
> > >
> > > **Response to Q2:** We greatly appreciate the Reviewer's question. We have exlained how to achieve the second objective in **Response to W3**. As you suggested, we replace $\theta$ using $\theta_o$, which makes the introduction from lines 256 to 268 more clear. We do not repeat the response to W3 here as the limitation of characters, please see the explanation in **Response to W3**.
> > >
> > >
> > >
> > >
> > >
> > > **Response to Q3:** We sincerely thank the Reviewer's question. In our continual learning scenario setting, it is reasonable that the unlearning user has some clean data to compose the $D_a$ and $D_c$. In the MLaaS scenarios, a huge amount of data is generated daily, and users may always have new data to update the model promptly. We have presented the user's ability in the above scenarios discussion. For the unlearning user, it would not upload $D_c$ that contains similar features to the unlearned data, as the user has the unlearning requirement and knows the unlearned data. If other users upload clean data that has features similar to those of unlearned data, it would not significantly impact the model's performance. We conducted additional experiments based on CIFAR10, as shown in the following table. The results demonstrate that mixing the unlearned samples into the constructed uploaded data for incremental learning negatively impacts the unlearning effect, as reflected by the increasing backdoor accuracy, but the model utility keeps.
> > >
> > >
> > > The **Tab.R1** of evaluating mixing unlearned data in the clean dataset on CIFAR10: USR=1%, ASR=1%, and CSR=1%:
> > >
> > > | On CIFAR10       | Mixed 0% of Unlearned Data, |  2%  | 4%  |  6% | 8% |
> > > | --------              | --------    | -------- | -------- |  -------- |   -------- |
> > > | Model Acc.       | 73.89%      | 73.85%   |  73.78%  |  73.25%   |  73.03%    |
> > > | Backdoor Acc.  | 9.40%       | 13.60%   | 33.40%   |  35.40%   |  43.26%    |
> > > | Running Time     | 6.63        | 6.72     | 6.83     |  7.01     |  7.16      |

---

> > > > ### Author Response · Authors · 2024-11-25
> > > > **Reminder on follow-up discussion (one day left before rebuttal ends)**
> > > >
> > > > Dear Reviewer BZGC,
> > > >
> > > > We sincerely appreciate the time and effort you've invested in reviewing our manuscript. Your expertise and dedication to the review process are truly invaluable to us.
> > > >
> > > > As the discussion phase draws to a close in one day, we kindly request you to provide your feedback on our rebuttal. Your insights are of immense importance to us, and we look forward to any additional comments you may have. If our replies have addressed your concerns, we would be thankful for your recognition of this. Should there be any points requiring further discussion or explanation, please feel free to contact us. Furthermore, we are fully prepared to engage more to enhance our work during this pivotal stage.
> > > >
> > > > Best regards,
> > > >
> > > > Authors

---

> > > > > ### Author Response · Authors · 2024-12-01
> > > > > **Reminder on follow-up discussion (one day left before the rebuttal extension ends)**
> > > > >
> > > > > Dear Reviewer BZGC,
> > > > >
> > > > > We sincerely appreciate the time and effort you've invested in reviewing our manuscript. Your expertise and dedication to the review process are truly invaluable to us.
> > > > >
> > > > > As the extended discussion phase draws to a close in one day, we kindly request you to provide your feedback on our rebuttal. Your insights are of immense importance to us, and we look forward to any additional comments you may have. If our replies have addressed your concerns, we would be thankful for your recognition of this. Should there be any points requiring further discussion or explanation, please feel free to contact us. Furthermore, we are fully prepared to engage more to enhance our work during this pivotal stage.
> > > > >
> > > > > Best regards,
> > > > >
> > > > > Authors

---

### Author Response · Authors · 2024-11-20
**Author Rebuttal by Authors**

# Rebuttal

Dear Reviewers, ACs, and PCs:

We are glad to receive valuable and constructive comments from all the reviewers. We have made a substantial effort to clarify reviewers' doubts and enrich our experiments in the rebuttal phase. In our responses, **Table R**xx or **Figure R**xx refers to the new **R**ebuttal results in the supplementary materials. We have uploaded the revised manuscript according to all reviewers' suggestions. We put the new experimental results and discussion at the end of the revised manuscript. We also provided supplementary materials, which separate the newly added context for your convenience. Below is a summary of our response:


**Reviewer BZGC:**

1. We revised the paper according to your detailed feedback in the weaknesses.

2. We clarified our notation by using $\theta_o$ to denote the original trained model and provided additional explanations for Equations (4) and (8).

3. We expanded the discussion on the threat model, detailing the abilities of both the server and the users in the unlearning scenario, and included relevant references to support our context.

4. We investigated the influence of when the clean dataset has similar features to the unlearned data and supplemented by additional experiments (**Tab. R1**), demonstrating that mixing unlearned data into the clean dataset negatively impacts the unlearning effect.


**Reviewer iCGz:**

1. We clarified the practical setting of our scenario by providing a discussion on the threat model, detailing the abilities of both the server and the users in the unlearning scenario, and including relevant references to support our context.

2. We explained how our Efficient Unlearning Update Estimation (EUUE) method works without requiring access to the training dataset, relying only on knowledge of the learning algorithm and some clean samples from the unlearning user. We also provided experimental results showing the scalability of our method of different learning rates (see **Tab. R2**).

3. We presented the original auxiliary data, their noisy counterparts, and directly constructed data without auxiliary data using [R11] in **Figure R1**.



**Reviewer 5Whi:**

1. We conducted additional experiments using the membership inference (MI) attack metric as you suggested, based on the referenced papers [R12] and [R13]. The results (see the **Tab.R3**) also demonstrate the effectiveness of OUbL.

2. We clarified that our approach does not require access to the training set and provided a detailed threat model.

3. We revised the presentation issues according to your suggestions. And we discussed potential solutions to mitigate the model utility degradation caused by unlearning.


**Reviewer EoYb:**

1. We clarified the scenario of our setting by providing a detailed discussion about the scenarios and threat model.

2. We added a discussion about the impact on the data distribution when unlearning occurs, and provided relevant references to support our context.

3. In response to your question about Figure 3, we clarified that the epsilon values of local differential privacy (LDP) pertain to the baseline VBU-LDP method, not our OUbL method. We revised the corresponding introduction for better clarity, explaining that the experiments aim to compare how much privacy protection our OUbL method achieves relative to the baseline unlearning method with a general LDP mechanism.

4. We also revised the presentation according to your suggestions.


**Reviewer Ha5f:**

1. We provided a detailed discussion of the threat model, outlining the abilities of both the ML server and the users.

2. We provided additional discussion about the practicality of our method and discussed potential solutions by constructing shadow models through hyperparameter stealing methods for a stricter practical setting.

3. We supplemented our paper with additional experiments demonstrating the computational efficiency of our method on MNIST, CIFAR10, and CelebA datasets in **Tab. R4**. The results show that even with the shifted computational costs, our method is more efficient than baseline methods like BFU, making it practical for real-world applications.


**Reviewer Gp18:**

1. We enhanced our manuscript by adding a detailed discussion of the scenario and threat model.

2. We conducted additional experiments with different learning rates to demonstrate the applicability of our method under various settings. The results show that a larger learning rate speeds up convergence and improves the unlearning effect, though it may slightly decrease model utility (see the **Tab.R2**).

3. We provided a detailed comparison between our method and existing unlearning works [R19, R20, R21], which you mentioned.

4. We conducted additional experiments on a tabular dataset Adult, and the results demonstrated the extensibility of our method, which are presented in **Tab.R5**.

We summarize new references used in rebuttal as follows.

---

> ### Author Response · Authors · 2024-11-20
> **References used in authors' response**
>
> [R1]. Hu, Hongsheng, et al. "Learn what you want to unlearn: Unlearning inversion attacks against machine unlearning." IEEE Symposium on Security and Privacy (SP) (2024).
>
> [R2]. Chen, Min, et al. "When machine unlearning jeopardizes privacy." Proceedings of the 2021 ACM SIGSAC conference on computer and communications security. 2021.
>
> [R3]. Rolnick, David, et al. "Experience replay for continual learning." Advances in neural information processing systems 32 (2019).
>
> [R4]. Lopez-Paz, David, and Marc'Aurelio Ranzato. "Gradient episodic memory for continual learning." Advances in neural information processing systems 30 (2017).
>
> [R5]. Wu, Yue, et al. "Large scale incremental learning." Proceedings of the IEEE/CVF conference on computer vision and pattern recognition. 2019.
>
> [R6]. Wang, Zifeng, et al. "Learning to prompt for continual learning." Proceedings of the IEEE/CVF conference on computer vision and pattern recognition. 2022.
>
> [R7]. Naseri, Mohammad, Jamie Hayes, and Emiliano De Cristofaro. "Local and central differential privacy for robustness and privacy in federated learning." NDSS (2022).
>
> [R8]. Bonawitz, Keith, et al. "Practical secure aggregation for privacy-preserving machine learning." proceedings of the 2017 ACM SIGSAC Conference on Computer and Communications Security. 2017.
>
> [R9]. Gao, Xiangshan, et al. "Verifi: Towards verifiable federated unlearning." IEEE Transactions on Dependable and Secure Computing (2024).
>
> [R10]. Guo, Xintong, et al. "Fast: Adopting federated unlearning to eliminating malicious terminals at server side." IEEE Transactions on Network Science and Engineering (2023).
>
>
> [R11]. Zhu, Ligeng, Zhijian Liu, and Song Han. "Deep leakage from gradients." Advances in neural information processing systems 32 (2019).
>
>
> [R12]. Kurmanji, M., Triantafillou, P., Hayes, J. and Triantafillou, E., 2024. Towards unbounded machine unlearning.Advances in neural information processing systems, 36.
>
> [R13]. Chen, Min, et al. "When machine unlearning jeopardizes privacy." Proceedings of the 2021 ACM SIGSAC conference on computer and communications security. 2021.
>
> [R14]. Hu, Hongsheng, et al. "Membership inference via backdooring." IJCAI (2022).
>
> [R15]. Bertram, Theo, et al. "Five years of the right to be forgotten." Proceedings of the 2019 ACM SIGSAC Conference on Computer and Communications Security. 2019.
>
> [R16]. Oh, Seong Joon, Bernt Schiele, and Mario Fritz. "Towards reverse-engineering black-box neural networks." Explainable AI: interpreting, explaining and visualizing deep learning (2019): 121-144.
>
> [R17]. Salem, Ahmed, et al. "{Updates-Leak}: Data set inference and reconstruction attacks in online learning." 29th USENIX security symposium (USENIX Security 20). 2020.
>
> [R18]. Wang, Binghui, and Neil Zhenqiang Gong. "Stealing hyperparameters in machine learning." 2018 IEEE symposium on security and privacy (SP). IEEE, 2018.
>
> [R19] Chundawat, Vikram S., et al. "Zero-shot machine unlearning." IEEE Transactions on Information Forensics andSecurity 18 (2023): 2345-2354.
>
> [R20] Golatkar, Aditya, Alessandro Achille, and Stefano Soatto. "Eternal sunshine of the spotless net: Selective forgettingin deep networks." Proceedings of the IEEE/CVF Conference on Computer Vision and Pattern Recognition. 2020.
>
> [R21] Tarun, Ayush K., et al. "Fast yet effective machine unlearning." IEEE Transactions on Neural Networks and LearningSystems (2023).

---

### Meta-Review · Area_Chair_8dd8 · 2024-12-10

**Metareview:**

- This paper addresses the challenge of machine unlearning without exposing erased data, providing an extra layer of privacy protection. It introduces the Oblivious Unlearning by Learning (OUbL) approach, where users synthesize a noisy auxiliary dataset to achieve unlearning through the model’s incremental learning process. Extensive experiments on various datasets and benchmarks demonstrate OUbL’s superior performance in privacy preservation and unlearning effectiveness.

- This paper is well-structured and easy to follow. The authors identify the privacy threats posed by the exposure of unlearned data during machine unlearning processes and propose a novel approach to achieve unlearning without exposing erased data.

- The main weaknesses are: (1) the scenario and threat model need more clarification; (2) the approach lacks theoretical guarantees regarding the privacy of users after unlearning; (3) it is unclear if it is feasible in a federated learning scenario; (4) there is an unrealistic assumption about the user's capability; and (5) the comparison with existing methods is not clear.

**Additional Comments On Reviewer Discussion:**

- The reviewers point out that the threat model and practical scenario are not clear, but the authors, during the rebuttal, sufficiently addressed these concerns by providing a continual learning scenario.
- The theoretical analysis is still missing even after the rebuttal.
- The computational cost analysis has also clearly addressed the reviewers' concerns.
- The comparison with existing methods is provided in detail.
- The concerns raised by Reviewer Ha5f have not been properly addressed. The authors claim that they have conducted an evaluation demonstrating the effectiveness of the proposed method under black-box assumptions. However, there are additional assumptions that might be hidden (e.g., the distribution of the data). Hence, it is not appropriate to claim that the method is effective under black-box assumptions. Moreover, Reviewer Ha5f's concern about the fact that updates from individual clients are typically aggregated on the server side, which significantly reduces the influence of any single client’s contributions, does not seem to have been addressed.

---

### Decision · Program_Chairs · 2025-01-22

Reject